# SafeDICE: Offline Safe Imitation Learning with Non-Preferred Demonstrations

**Youngsoo Jang[1], Geon-Hyeong Kim[1], Jongmin Lee[2],**
**Sungryull Sohn[1], Byoungjip Kim[1], Honglak Lee[1], Moontae Lee[1,3]**
[1] LG AI Research
[2] University of California, Berkeley
[3] University of Illinois Chicago

## Abstract

We consider offline safe imitation learning (IL), where the agent aims to learn the safe policy that mimics preferred behavior while avoiding non-preferred behavior from non-preferred demonstrations and unlabeled demonstrations. This problem setting corresponds to various real-world scenarios, where satisfying safety constraints is more important than maximizing the expected return. However, it is very challenging to learn the policy to avoid constraint-violating (i.e. non-preferred) behavior, as opposed to standard imitation learning which learns the policy to mimic given demonstrations. In this paper, we present a hyperparameter-free offline safe IL algorithm, SafeDICE, that learns safe policy by leveraging the non-preferred demonstrations in the space of stationary distributions. Our algorithm directly estimates the stationary distribution corrections of the policy that imitate the demonstrations excluding the non-preferred behavior. In the experiments, we demonstrate that our algorithm learns a more safe policy that satisfies the cost constraint without degrading the reward performance, compared to baseline algorithms.

## 1 Introduction

Reinforcement learning (RL) [28] aims to learn the optimal policy that maximizes the expected cumulative rewards through interacting with the environment. Recently, RL has achieved remarkable successes in various domains, but its application to real-world tasks is yet challenging mainly due to the following two reasons: First, since the interaction with the environment is costly in most real-world tasks, the policy should be learned solely from the pre-collected dataset (i.e. *offline learning*). Second, reward annotation, which is essential for applying the RL algorithm, is a very strong requirement that is difficult to obtain in real-world tasks.

Imitation learning (IL) [26, 1, 25] is a more realistic framework to deal with real-world tasks, where the expert's demonstrations are solely given without reward annotations. Standard IL aims to mimic the expert's demonstrations generated from the optimal policy in terms of expected return. However, in most real-world tasks, additional constraints should be considered in addition to maximizing the expected return such as the safe driving for autonomous driving and toxicity degree for the conversational agent. Fortunately, *non-preferred* demonstrations that violate constraints are often naturally collected. For example, black box data of an accident vehicle and toxic content reported by chatbot users are separately collected as *non-preferred* demonstrations. However, most existing studies have focused only on imitating expert (i.e. *preferred*) behavior, and a safe imitation learning method that avoids *non-preferred* behavior has not yet been explored.

In this paper, we are particularly interested in solving safe imitation learning in an offline setting: Given scarce but labeled non-preferred demonstrations, and abundant but unlabeled demonstrations

37th Conference on Neural Information Processing Systems (NeurIPS 2023).

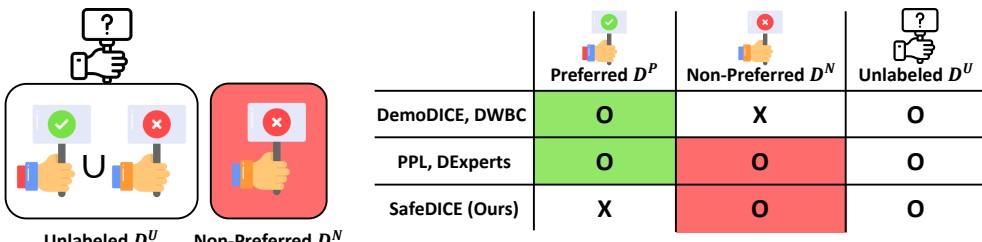

| | Preferred $D^P$ | Non-Preferred $D^N$ | Unlabeled $D^U$ |
|---|---|---|---|
| DemoDICE, DWBC | O | X | O |
| PPL, DExperts | O | O | O |
| SafeDICE (Ours) | X | O | O |

Figure 1: Illustrative example of our problem setting and the position of work among the prior methods and problem settings. First, prior offline imitation learning algorithms (DemoDICE [13], DWBC [31]), which aim to imitate expert demonstrations, assume labeled preferred demonstrations and unlabeled demonstrations. Second, preference-based policy learning (PPL) algorithms and DExperts [19] require both labels of preferred and non-preferred demonstrations. Unlike the problem settings of prior works, our work focuses on the problem of safe imitation learning assuming only the labels of non-preferred demonstration that can deal with safety more importantly.

of both preferred and non-preferred behavior, we aim to learn a safe policy that follows the preferred behavior while avoiding the non-preferred behavior (see Figure 1). This problem setting has been mostly unexplored, and cannot be completely addressed by existing methods. One of the existing applicable approaches is the preference-based policy learning algorithm [4, 35, 17], which optimizes a policy based on preference rankings over demonstrations. This method can learn the policy to prefer unlabeled demonstrations to non-preferred demonstrations, but it also prefers non-preferred demonstrations mixed in the unlabeled demonstrations, which can lead to learning suboptimal policy. One of the approaches effectively using the unlabeled dataset is DWBC [31], a weighted behavior cloning based on a discriminator, which adopts the positive-unlabeled learning [9, 36, 30] for training the discriminator. This method requires an additional cost of hyperparameter search for training the discriminator, and its performance is hyperparameter sensitive. Another notable approach in the natural language processing domains is DExperts [19], a decoding-time method, which combines trained models with preferred and non-preferred demonstrations. This method is also hyperparameter sensitive and prone to the *degeneration* issue that leads to abnormal behavior. Both DWBC and DExperts, which require hyperparameter search, are unsuitable for real-world tasks where interaction with the real environment is costly and risky.

We present an offline safe IL algorithm that learns safe policy by leveraging the non-preferred demonstrations in the space of stationary distributions. Our algorithm, *offline Safe imitation learning by avoiding non-preferred demonstrations via DIstribution Correction Estimation* (SafeDICE), directly estimates the stationary distribution corrections of the policy that imitate the demonstrations excluding the non-preferred behavior. SafeDICE essentially leverages the non-preferred demonstrations in the space of stationary distributions, unlike existing algorithms that leverage it in the space of policy [19] or the learning process of the discriminator [31]. We show that our algorithm can be reduced to a single convex minimization problem *without additional hyperparameter search*, unlike existing algorithms that are highly sensitive to the hyperparameter. In the experiments, we demonstrate that our algorithm learns a more preferred (i.e. constraint-satisfying) policy compared to baseline algorithms in constrained RL benchmarks including RWRL [8] and Safety Gym [24].

## 2 Preliminaries

### 2.1 Markov Decision Process (MDP)

We consider an environment modeled as a Markov Decision Process (MDP), defined by $M = \langle S, A, T, R, p_0, \gamma \rangle$ [28], where $S$ is the set of states, $A$ is the set of actions, $T : S \times A \to \Delta(S)$ is the transition probability, $R : S \times A \to \mathbb{R}$ is the reward function, $p_0 \in \Delta(S)$ is the distribution of the initial state, and $\gamma \in [0, 1)$ is the discount factor. For the given policy $\pi$, its stationary distribution $d^\pi(s, a)$ is defined as follows:

$$d^\pi(s, a) := (1 - \gamma) \sum_{t=0}^{\infty} \gamma^t \Pr(s_t = s, a_t = a)$$

where $s_0 \sim p_0$, $a_t \sim \pi(\cdot|s_t)$, and $s_{t+1} \sim T(\cdot|s_t, a_t)$ for all timesteps $t \geq 0$. For brevity, $d^\pi$ will be used to denote the stationary distribution $d^\pi(s, a)$.

We consider the offline imitation learning setting, where the policy should be optimized from solely a pre-collected dataset without reward annotation and any additional online environment interaction. In our problem setting, *preferred* and *non-preferred* behaviors are defined by satisfying or violating constraints. We denote the preferred policy by $\pi^P$, the non-preferred policy by $\pi^N$, and the corresponding stationary distribution of $\pi^P$ and $\pi^N$ by $d^P$ and $d^N$, respectively. We assume a pre-collected dataset that consists of *scarce but labeled* dataset $D^N$ generated by the non-preferred policy, and a *abundant but unlabeled* dataset $D^U$ generated by both preferred and non-preferred policies. We denote the corresponding stationary distribution of $D^U$ by $d^U$, which can be regarded as convex combination of $d^P$ and $d^N$ with unknown ratio $\alpha$ (i.e. $d^U := (1 - \alpha)d^P + \alpha d^N$). In this setting, we aim to learn a *safe* policy from unlabeled demonstrations while *avoiding non-preferred* demonstrations that are labeled as non-preferred behavior.

## 2.2 Preference-based Offline Imitation Learning

Imitation learning (IL) aims to imitate the behavior of expert policy solely from state-action demonstrations without reward annotations. The standard IL is naturally formulated as minimizing the KL divergence between stationary distributions of preferred policy (i.e. expert policy) and target policy [14, 11]:

$$\pi^* := \arg\min_\pi D_{\mathrm{KL}}(d^\pi \| d^P). \tag{1}$$

However, it requires a large amount of annotated preferred demonstrations, which may be a too strong requirement for various real-world scenarios. Recent offline IL method [13] relaxes this requirement for preferred demonstrations by leveraging the unlabeled dataset which may contain both preferred and non-preferred demonstrations:

$$\pi^* := \arg\min_\pi D_{\mathrm{KL}}(d^\pi \| d^P) + \alpha D_{\mathrm{KL}}(d^\pi \| d^U), \tag{2}$$

where $\alpha \geq 0$ is a hyperparameter that controls the balance between minimizing the KL divergence with $d^U$ and $d^P$. This offline imitation learning method can learn the preferred policy from a large number of unlabeled demonstrations with only a small amount of labeled preferred demonstrations.

In contrast to the mainstream of recent IL approaches focusing on mimicking expert (i.e. preferred) behavior, avoiding dangerous (i.e. non-preferred) behavior is a more important problem in many real-world tasks, which has been unexplored yet. In the same manner as Eq. (2), we can naively consider the penalty for the non-preferred behavior by maximizing the KL divergence with $d^N$:

$$\pi^* := \arg\min_\pi -D_{\mathrm{KL}}(d^\pi \| d^N) + \alpha D_{\mathrm{KL}}(d^\pi \| d^U), \tag{3}$$

where $\alpha \geq 0$ is a hyperparameter that controls the balance between minimizing the KL divergence with $d^U$ and maximizing the KL divergence with $d^N$. However, since the KL divergence has no upper bound, optimizing Eq. (3) is very unstable and challenging, where the $D_{\mathrm{KL}}(d^\pi \| d^N)$ can blow up to infinity. Therefore, safe imitation learning while avoiding non-preferred behavior is not a straightforward problem and has not yet been studied.

## 3 SafeDICE

In this section, we present *offline Safe imitation learning by avoiding the non-preferred demonstrations via DIstribution Correction Estimation* (SafeDICE), an offline IL algorithm that can learn the safe policy using labeled non-preferred demonstrations. SafeDICE optimizes the stationary distributions of the target policy to match the stationary distribution of the preferred policy from the unlabeled dataset by excluding non-preferred demonstrations.

### 3.1 Offline Safe Imitation Learning with Non-Preferred Demonstrations

Similar to standard IL approaches, the derivation of our algorithm starts from KL divergence minimization between $d^\pi$ and $d^P$, which can be estimated by $d^U$ and $d^N$ as follows:

$$\pi^* = \arg\min_\pi D_{\mathrm{KL}}\left(d^\pi \,\middle\|\, \frac{d^U - \alpha d^N}{1 - \alpha}\right), \tag{4}$$

where $\alpha$ is *unknown* ratio between the stationary distributions of preferred policy and non-preferred policy (i.e. $d^U := (1-\alpha)d^P + \alpha d^N$). Optimizing the Eq. (4) is sensitive to $\alpha$, and selecting the $\alpha$ with hyperparameter search is not suitable for real-world tasks where the interaction with the real environment is very costly and risky. Therefore, it is not straightforward to learn the preferred policy using non-preferred demonstrations with existing imitation learning algorithms. The main contribution of our paper is to present an algorithm that theoretically determines $\alpha$ and learns the preferred policy only with the labels of non-preferred demonstrations without any hyperparameter search. In the following description, we first consider the $\alpha$ as a known ratio in our derivation, then present the way to select the $\alpha$ while providing the theoretical guarantees on the error bound of optimized stationary distribution with selected $\alpha$. All the proofs can be found in Appendix A.

The main flow of our derivation follows prior DICE-based offline imitation learning methods [12, 13], but it is clearly different from the existing methods that require hyperparameter search to deal with unknown $\alpha$. First, we consider a problem equivalent to Eq. (4) in terms of stationary distribution $d$:

$$\max_{d \geq 0} \quad -D_{\mathrm{KL}}\left(d \,\middle\|\, \frac{d^U - \alpha d^N}{1-\alpha}\right) \tag{5}$$

$$\text{s.t. } (\mathcal{B}_* d)(s) = (1-\gamma)p_0(s) + \gamma(\mathcal{T}_* d)(s) \quad \forall s, \tag{6}$$

where $(\mathcal{B}_* d)(s) := \sum_a d(s,a)$ is the marginalization operator, and $(\mathcal{T}_* d)(s) := \sum_{\bar{s},\bar{a}} T(s|\bar{s},\bar{a})d(\bar{s},\bar{a})$ is the transposed Bellman operator. The Bellman flow constraint (6) ensures that $d(s,a)$ is a valid stationary distribution of some policy, where $d(s,a)$ can be interpreted as a normalized occupancy measure of $(s,a)$. Therefore, if we find optimal $d^*(s,a)$ that satisfies Eq. (5) and (6), then its corresponding policy can be easily obtained by $\pi^*(a|s) = \frac{d^*(s,a)}{\sum_a d^*(s,a)}$.

The Lagrangian dual formulation of the above constrained optimization problem is

$$\max_{d \geq 0} \min_{\nu} -D_{\mathrm{KL}}\left(d \,\middle\|\, \frac{d^U - \alpha d^N}{1-\alpha}\right) + \sum_s \nu(s)\Big((1-\gamma)p_0(s) + \gamma(\mathcal{T}_* d)(s) - (\mathcal{B}_* d)(s)\Big), \tag{7}$$

where $\nu(s)$ are the Lagrange multipliers for the Bellman flow constraints (6). However, directly optimizing the Eq. (7) is challenging in an offline manner, where further environment interactions are not available. To obtain a tractable objective, we introduce the following derivations:

$$-D_{\mathrm{KL}}\left(d \,\middle\|\, \frac{d^U - \alpha d^N}{1-\alpha}\right) + \sum_s \nu(s)\Big((1-\gamma)p_0(s) + \gamma(\mathcal{T}_* d)(s) - (\mathcal{B}_* d)(s)\Big)$$

$$= (1-\gamma)\mathbb{E}_{p_0}[\nu(s)] + \mathbb{E}_d\left[\gamma(\mathcal{T}\nu)(s,a) - \nu(s) - \log\frac{(1-\alpha)d(s,a)}{d^U(s,a) - \alpha d^N(s,a)}\right]$$

$$= (1-\gamma)\mathbb{E}_{p_0}[\nu(s)] + \mathbb{E}_d\left[\gamma(\mathcal{T}\nu)(s,a) - \nu(s) - \log\underbrace{\frac{d(s,a)}{d^U(s,a)}}_{:=w(s,a)} + \log\underbrace{\frac{d^U(s,a) - \alpha d^N(s,a)}{(1-\alpha)d^U(s,a)}}_{:=r(s,a)}\right]$$

$$= (1-\gamma)\mathbb{E}_{p_0}[\nu(s)] + \mathbb{E}_d\left[\underbrace{r_\alpha(s,a) + \gamma(\mathcal{T}\nu)(s,a) - \nu(s)}_{:=A_\nu(s,a)} - \log w(s,a)\right]$$

$$= (1-\gamma)\mathbb{E}_{p_0}[\nu(s)] + \mathbb{E}_{d^U}\left[w(s,a)\big(A_\nu(s,a) - \log w(s,a)\big)\right]$$

$$=: \mathcal{L}(w,\nu; r_\alpha),$$

where $r_\alpha(s,a)$ corresponds to the preference-based reward function in the context of RL, which will be trained by the pre-collected dataset.

In summary, our goal is reduced to solve the following maximin optimization:

$$\max_{w \geq 0} \min_{\nu} \mathcal{L}(w,\nu; r_\alpha), \tag{8}$$

where the optimal solution $w^*$ of Eq. (8) represents the stationary distribution correction of an optimal policy $\pi^*$ (i.e. $w^*(s,a) = \frac{d^{\pi^*}(s,a)}{d^U(s,a)}$).

### 3.2 Pretraining the Preference-based Reward Function $r_\alpha(s, a)$

To optimize the Eq. (8), we pretrain the discriminator to estimate the log ratio of $d^N(s, a)$ and $d^U(s, a)$, which is required to estimate for the $r_\alpha(s, a)$:

$$c^* = \arg\max_c \mathbb{E}_{d^N}[\log c(s, a)] + \mathbb{E}_{d^U}[\log(1 - c(s, a))]. \tag{9}$$

With the optimal discriminator $c^*(s, a) = \frac{d^N(s,a)}{d^U(s,a)+d^N(s,a)}$ trained by Eq. (9), the preference-based reward function $r_\alpha(s, a)$ can also be obtained as:

$$r_\alpha(s, a) = \log\left(\frac{1 - (1 + \alpha)c^*(s, a)}{(1 - \alpha)(1 - c^*(s, a))}\right). \tag{10}$$

However, according to the value of $\alpha$, the reward function $r_\alpha(s, a)$ defined by Eq. (10) cannot guarantee that the value inside the log function always becomes a positive. Therefore, finding the optimal $\alpha$ through hyperparameter search is infeasible.

### 3.3 Reduction to the Single Min Optimization

Since the optimization Eq. (7) is a convex optimization, we can change the order of operator from maximin optimization to minimax optimization [2]:

$$\min_\nu \max_{w \geq 0} \mathcal{L}(w, \nu; r). \tag{11}$$

Then, we can replace the inner maximization of Eq. (11) with a closed-form solution for $w$, which can be easily obtained by exploiting the convexity.

**Proposition 3.1.** *For any $\nu$, the closed-form solution to the inner maximization of Eq. (11), i.e. $w_\nu^* = \arg\max_{w \geq 0} \mathcal{L}(w, \nu; r)$, is given by:*

$$w_\nu^*(s, a) = \exp(A_\nu(s, a) - 1). \tag{12}$$

By using the closed-form solution $w_\nu^*$ of Eq. (12), our derivation can be reduced to a single minimization as follows:

$$\min_\nu \mathcal{L}(w_\nu^*, \nu; r) = (1 - \gamma)\mathbb{E}_{p_0}[\nu(s)] + \mathbb{E}_{d^U}\left[\exp(A_\nu(s, a) - 1)\right]. \tag{13}$$

Finally, by optimizing the single minimization problem of Eq. (13), we can obtain the optimal stationary distribution correction $w^*$ which can easily extract the corresponding optimal policy $\pi^*$. To extract a policy from the optimal stationary distribution correction $w^*(s, a)$, we adopt weighted behavior cloning on the offline dataset as follows:

$$\min_\pi -\mathbb{E}_{(s,a)\sim d^{\pi^*}}[\log \pi(a|s)] = -\mathbb{E}_{(s,a)\sim d^U}[w^*(s, a)\log \pi(a|s)], \tag{14}$$

where $w^*(s, a) = \frac{d^{\pi^*}(s,a)}{d^U(s,a)}$.

### 3.4 Hyperparameter $\alpha$ Selection

In the previous sections, we have derived our algorithm under the assumption that $\alpha$ is a known ratio. The remaining question is how we select the suitable $\alpha$ without additional hyperparameter search. Fortunately, in contrast to the existing methods [19, 31] that require the additional cost of hyperparameter search, we can specify the $\alpha$ with providing the theoretical guarantees on the error bound of the estimated stationary distribution.

**Proposition 3.2.** *Suppose $\min_{(s,a)} \frac{d^P(s,a)}{d^N(s,a)} \leq \epsilon$. If we obtain the estimated stationary distribution by optimizing the Eq. (7) with $\alpha$ set to $\hat{\alpha}$ as follows:*

$$\hat{\alpha} = \min_{(s,a)} \frac{d^U(s, a)}{d^N(s, a)}, \tag{15}$$

*then the total variation distance of true and estimated stationary distribution of preferred policy is bounded as follows:*

$$D_{\mathrm{TV}}(d^P(s,a), \hat{d}^P(s,a)) \leq \frac{\epsilon}{1+\epsilon}, \tag{16}$$

*where $D_{\mathrm{TV}}$ is total variation distance, $d^P(s,a)$ is true stationary distribution of the preferred policy and $\hat{d}^P(s,a)$ is estimated stationary distribution which is defined by $\hat{\alpha}$ as $\hat{d}^P(s,a) = \frac{d^U(s,a) - \hat{\alpha} d^N(s,a)}{1 - \hat{\alpha}}$. (Proof in Appendix A.)*

Intuitively, since the behavior of the non-preferred and preferred policies are different, there must exist state-action spaces that are rarely visited by the preferred policy but are frequently visited by the non-preferred policy. Therefore $\epsilon$ may be a very small positive value, then the total variation distance of the true and estimated stationary distribution is close to zero:

$$\lim_{\epsilon \to 0} D_{\mathrm{TV}}(d^P(s,a), \hat{d}^P(s,a)) = 0. \tag{17}$$

We also provide proof in Appendix A.3 that $r_\alpha(s,a)$ of Eq. (10), defined by the $\alpha$ selected by Eq. (15), is always well-defined with positive value inside the log-function in Eq. (10).

Finally, our algorithm optimizes the Eq. (7) with the $\alpha$ obtained by Eq. (15), where $\alpha$ is no longer an unknown hyperparameter. In short, by leveraging the non-preferred demonstrations in the space of stationary distributions, this problem can be resolved by *solving a single convex minimization problem for a specific $\alpha$*. This is in contrast to the existing methods, which directly leverage the non-preferred demonstrations for training the policy [19] or discriminator [31] that require the additional cost of hyperparameter search. The pseudocode for the whole process of SafeDICE can be found in Appendix B.

## 4 Related Work

**Stationary distribution correction estimation (DICE)** Since the introduction of DICE for off-policy evaluation in DualDICE [21], various DICE-family algorithms have been developed for off-policy evaluation [33, 34, 32, 7]. In addition to these studies, two algorithms, AlgaeDICE [22] and ValueDICE [14], have been proposed to address the challenges of off-policy RL and IL, respectively. However, they may encounter issues with numerical instability in an offline setting due to the nested min-max optimization and out-of-distribution action evaluation.

To overcome the issues of numerical instability in offline settings, recent DICE-based offline RL [15, 16] and IL algorithms [13, 20, 12] have adopted a strategy of directly optimizing stationary distribution correction ratios, similar to the approach used in our work. Additionally, these offline IL algorithms utilize unlabeled demonstrations in an effort to address the problem of distribution drift [25]. They share a similarity with our algorithm in that they all aim to recover the expert's behavior through direct optimization of the stationary distribution correction ratios. However, these algorithms rely on the availability of clean expert demonstrations, which can make them difficult to use in situations where expert demonstrations are not available.

**Policy Learning from Suboptimal Demonstrations** Preference-based policy learning algorithms use ranked suboptimal demonstrations to learn a reward function. Many of these algorithms, such as T-REX [4], PrefPPO [18], and PEBBLE [17], utilize Bradley-Terry model [3] to model the reward function. Once the reward function is learned, T-REX and PrefPPO use PPO [27] to train a policy, while PEBBLE employs SAC [10]. While preference-based policy learning algorithms allow for learning from unlabeled demonstrations and potentially disregarding the negative influence of non-preferred demonstrations, it is important to note that they may also give preference to non-preferred demonstrations mixed in the unlabeled demonstrations.

On the other hand, there are also many studies that utilize suboptimal demonstrations, even though they are not preference-based policy learning algorithms. D-REX [5] and SSRR [6] both learn a policy by using given suboptimal demonstrations, and then generate trajectories with varying levels of optimality using this policy with injected noise. But, both algorithms require online sampling, which is not feasible in offline settings. Wu et al. [29] proposed two algorithms that utilize positive-unlabeled learning [9, 36, 30] to train a discriminator for constructing a reward function. However,

both algorithms require ground-truth labels for uniformly sampled trajectories from unlabeled demonstrations. DWBC [31] uses positive-unlabeled learning to train a discriminator, which is then used to construct weights for weighted BC. DExperts [19], a decoding-time method in the natural language processing domain, combines models for preferred and non-preferred demonstrations. Unfortunately, both DWBC and DExperts may not be suitable for offline safe imitation learning tasks due to their sensitivity to hyperparameter settings.

# 5    Experiments

In this section, we show the experimental results on various tasks from constrained RL benchmarks. First, we conduct the evaluation on domains from Real-World RL (RWRL) suite [8], which provides challenges in real-world scenarios including safety constraints. Second, we also conduct the experiment on continuous control tasks from Safety Gym environment [24], which provides practical scenarios of safety issues. In order to show the motivation of our work and the limitation of existing algorithms, we also provide qualitative results on tabular MDP environments. The illustrative examples and qualitative analysis on tabular MDP environments can be found in Appendix G.

**Baselines** Since offline safe IL, which we consider in this paper, has not been explored yet much, it lacks directly applicable baseline algorithms. Therefore, we consider the following four algorithms as baselines. First, BC denotes simple behavior cloning on unlabeled demonstrations. Second, DWBC denotes a variant of the original discriminator-weighted behavior cloning algorithm [31] that adopts the positive-unlabeled learning for training the discriminator. To apply it to our setting, we modified positive-unlabeled learning to negative-unlabeled learning. Third, PPL denotes a preference-based policy learning algorithm that adopts the preference-based reward learning algorithm based on the Bradley-Terry model [3] and weighted behavior cloning for policy learning. Lastly, we also compare with DExperts [19], a decoding-time method that combines trained models with unlabeled and non-preferred demonstrations. The details of baseline algorithms are provided in Appendix C.

**Evaluation metrics** We compare the performance of SafeDICE and baseline algorithms in terms of task performance (i.e. expected return) and safety (i.e. cost constraints). We report the results with the following metrics: 1) *Normalized Return*: evaluates the normalized task performance of the policy, 2) *Average Cost*: evaluates the mean cost of the policy, 3) *Cost-Violating Ratio*: evaluates the ratio of cost-violating behavior of the policy, 4) *Conditional Value at Risk performance (CVaR) 10% Cost*: evaluates the mean cost of the worst 10% runs. We also provide *CVaR k% Cost* for various values of $k$ in Appendix E.1. Here, our main focus of the comparison is to evaluate the safety (*Average Cost*, *Cost-Violating Ratio*, and *CVaR 10% Cost*), and we use the *Normalized Return* to examine the degeneration issue, whether the agents fall into abnormal behaviors or not. All metrics are measured as the average value of 500 trajectories generated from the learned policy, and we report the results that are averaged over 5 runs.

## 5.1    Real-World Reinforcement Learning (RWRL) Suite

Now, we present the results of SafeDICE on a set of continuous control benchmarks from Real-World RL (RWRL) suite [8], where the safety constraints are also provided by environments.

**Task setup** We conduct the experiments for three tasks from RWRL environment with safety constraints: Cartpole, Walker, and Quadruped. Since there is no standard dataset for offline IL considering the safety constraints, we collected data by training the online RL agents. To obtain an preferred and non-preferred policy, we trained the SAC [10] in different two ways that solve the RL problem with respect to reward penalized by the cost or not. Among these two policies which are both optimal in terms of return maximization, we define the preferred policy as a policy that satisfies the constraint (i.e. high-return-low-cost behavior) and the non-preferred policy as a policy that violates the constraint (i.e. high-return-high-cost behavior). This experimental setting is representative of many real-world scenarios, where the data is collected by high-return behavior without consideration of the cost. Then, we generated scarce but labeled non-preferred demonstrations $D^N$ from the non-preferred policy, and abundant but unlabeled demonstrations $D^U$ from both preferred and non-preferred policies. We collected diverse trajectories by sampling with a stochastic policy from randomly initialized initial states. Our goal is to recover the low-cost behavior by avoiding the high-cost demonstrations while not hurting the high-return behavior of unlabeled demonstrations. The details for the experimental settings can be found in Appendix D.

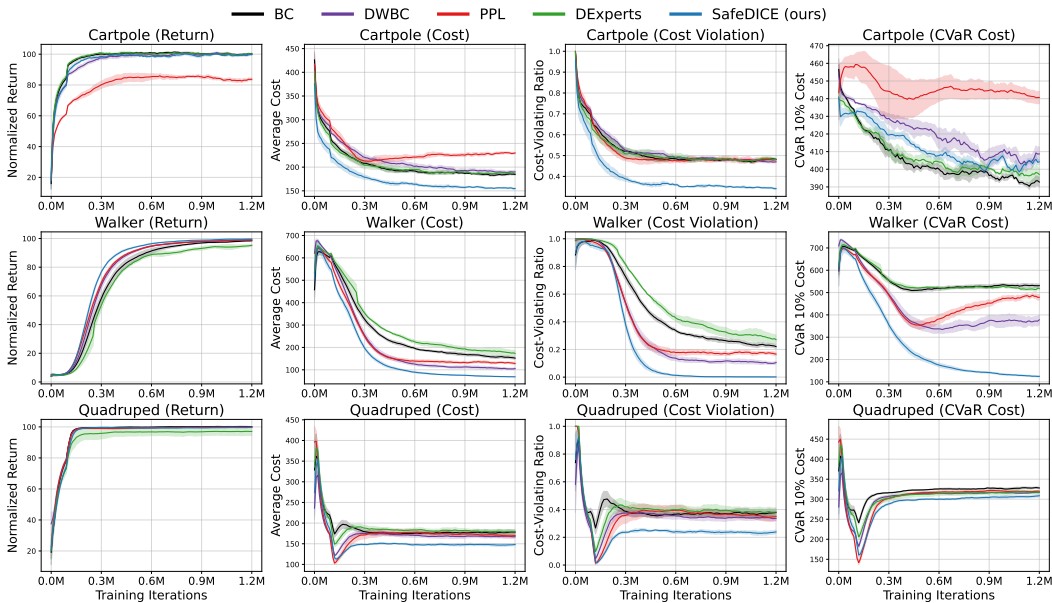

Figure 2: Experimental results on RWRL control tasks. All results are averaged over 5 runs, and the shaded area represents the standard error.

**Main results** Figure 2 summarizes the overall performance of SafeDICE and baseline algorithms in RWRL tasks. The results show that SafeDICE significantly outperforms baseline algorithms in all metrics related to safety across all domains (except for *CVaR 10% Cost* in the Cartpole domain), while not hurting the performance of normalized return. In the case of the Cartpole domain, since there are cases where the cost is unavoidably violated depending on the initial position of cart that is initialized randomly, *CVaR 10% Cost* which measures the worst 10% performance shows similar performance to other algorithms. We have also evaluated the *CVaR $k$% Cost* with various $k$ and are provided in Appendix E.1. PPL outperforms BC on Walker and Quadruped domains but suffers from degeneration issues on the Cartpole domain. DWBC outperforms or matches the other baseline algorithms on all domains, but performs poorly compared to SafeDICE despite the best results obtained through the additional hyperparameter search. Similar to DWBC, DExperts shows results that are sensitive to hyperparameters, and are vulnerable to degeneration issues depending on hyperparameters. Additional experimental results according to various hyperparameters for DWBC and DExperts are provided in Appendix E.2.

**Results with varying amount of $D^N$** To study the dependence on the amount of labeled non-preferred demonstrations, we experiment with different numbers of labeled non-preferred demonstrations $|D^N|$ on Walker domain. (The results are provided in Appendix E.3.) Figure 8 summarizes the performance of SafeDICE and baseline algorithms with varying the number of labeled non-preferred demonstrations ($|D^N| \in \{50, 20, 10, 5\}$). In this experiment, it is expected that performance in terms of safety gradually degrades as $|D^N|$ decreases. As shown in Figure 8, as $|D^N|$ decreases, SafeDICE shows a greater difference from other baseline algorithms, significantly outperforming. PPL works similarly to SafeDICE when $|D^N|$ is large but shows sensitive performance depending on $|D^N|$. DWBC is not sensitive to $|D^N|$, but it performs poorly compared to SafeDICE. DExperts easily suffers from degeneration issues when $|D^N|$ is small and performs poorly even when $|D^N|$ is large. These results represent that our algorithm SafeDICE can be applied more effectively in real-world scenarios where labeled non-preferred demonstrations are not provided in large quantities.

## 5.2 Safety Gym Environment

To show that our algorithm robustly works in various environments, we additionally conduct experiments on Safety Gym[24], which is one of the representative benchmarks for safety in RL.

**Task setup** We conduct the experiments for Goal and Button tasks from the Safety Gym environment. For each task, the agent aims to maximize task performance while satisfying the constraints by avoiding the dangerous area (see Figure 3). Since there is no standard dataset for offline IL considering

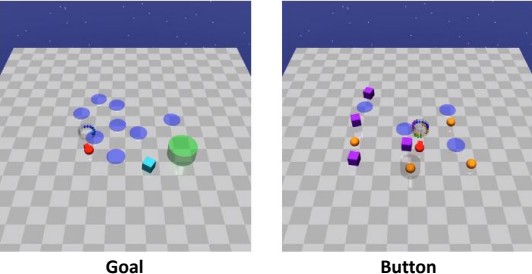

**Goal**        **Button**

Figure 3: Tasks of Safety Gym environment. For each task, the agent (red robot) aims to achieve the objective while satisfying the cost constraints. The agents receive costs when entering hazards indicated by blue circles or when touching the obstacles. The objectives of each task are as follows: **Goal:** Move to a series of green goal positions **Button:** Press a series of highlighted goal buttons.

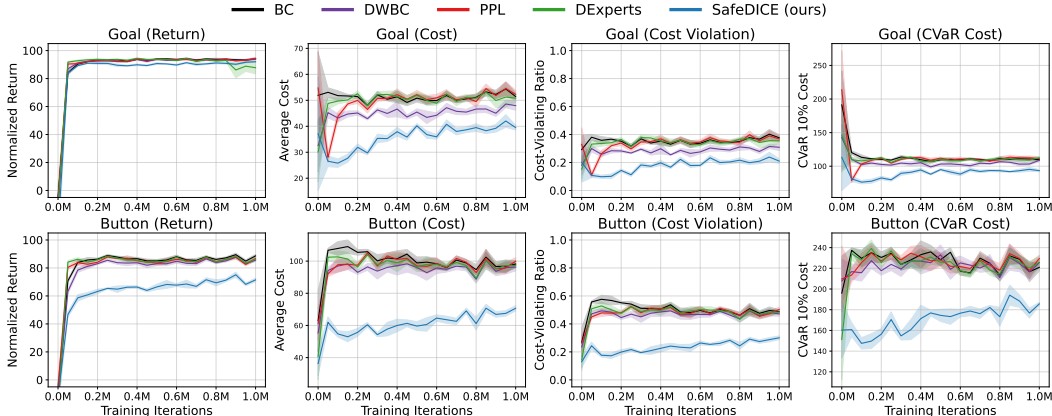

Figure 4: Experimental results on Safety Gym control tasks. All results are averaged over 5 runs, and the shaded area represents the standard error.

the safety constraints, we collected data by training the online RL agents. To obtain non-preferred and preferred policies, we trained PPO [27] and PPO-Lagrangian [24], which is combined with the Lagrangian approach to PPO, respectively. Here, we define the non-preferred policy as a policy that violates the constraint and the preferred policy as a policy that satisfies the constraint. Then, we generated scarce but labeled non-preferred demonstrations $D^N$ from the PPO-Lagrangian policy, and abundant but unlabeled demonstrations $D^U$ from both PPO and PPO-Lagrangian policies. We collected diverse trajectories by sampling with a stochastic policy from randomly initialized initial states. In this setting, our goal is to learn a safe policy (i.e. preferred policy) that satisfies the cost constraints by avoiding the cost-violating (i.e. non-preferred) demonstrations. The details for the experimental settings can be found in Appendix D.

**Results** Figure 4 summarizes the overall performance of SafeDICE and baseline algorithms in Safety Gym tasks. The results show that SafeDICE significantly outperforms baseline algorithms in all metrics related to safety across all domains. In the result of the Button task, SafeDICE shows low performance in *Return*, but it is a desirable result because the non-preferred demonstrations were collected from a safe policy with relatively low returns and very low costs (see Appendix D).

## 6 Conclusion

In this work, we presented SafeDICE, an offline safe IL algorithm, which can learn safe policy by leveraging the non-preferred demonstrations in the space of stationary distributions. We formulated this problem as a stationary distribution matching problem, which can be reduced to a single convex minimization problem without additional hyperparameter search, unlike existing algorithms that are highly sensitive to the hyperparameter. In the experiments, we demonstrated that SafeDICE learns a more safe policy that satisfies the cost constraint in constrained RL benchmarks including RWRL and Safety Gym. As the future work, we will consider an extension to the problem settings where preference information is given as a real value rather than binary value. There are no potential negative societal impacts of our work.

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

## A  Theoretical Analysis

### A.1  Closed-form solutions

**Proposition 3.1.** *For any $\nu$, the closed-form solution to the inner maximization of Eq.* (11)*, i.e.* $w_\nu^* = \arg\max_{w \geq 0} \mathcal{L}(w, \nu; r)$*, is given by:*

$$w_\nu^*(s, a) = \exp(A_\nu(s, a) - 1). \tag{12}$$

*Proof.* For $(s, a)$ with $d^U(s, a) > 0$, we can compute the derivative $\frac{\partial \mathcal{L}(w, \nu; r)}{\partial w(s, a)}$ of $\mathcal{L}(w, \nu; r)$ w.r.t. $w(s, a)$ as follows:

$$\frac{\partial \mathcal{L}(w, \nu; r)}{\partial w(s, a)} = \sum_{(s,a)} d^U(s, a) \left( A_\nu(s, a) - \log w(s, a) + w(s, a) \cdot -\frac{1}{w(s, a)} \right) = 0$$

$$\Leftrightarrow A_\nu(s, a) - \log w(s, a) - 1 = 0$$
$$\Leftrightarrow \log w(s, a) = A_\nu(s, a) - 1$$
$$\Leftrightarrow w(s, a) = \exp(A_\nu(s, a) - 1).$$

$\square$

### A.2  Theoretical analysis of error bound between $d^P$ and $\hat{d}^P$

**Proposition 3.2.** *Suppose* $\min_{(s,a)} \frac{d^P(s,a)}{d^N(s,a)} \leq \epsilon$*. If we obtain the estimated stationary distribution by optimizing the Eq.* (7) *with $\alpha$ set to $\hat{\alpha}$ as follows:*

$$\hat{\alpha} = \min_{(s,a)} \frac{d^U(s, a)}{d^N(s, a)}, \tag{15}$$

*then the total variation distance of true and estimated stationary distribution of preferred policy is bounded as follows:*

$$D_{\mathrm{TV}}(d^P(s, a), \hat{d}^P(s, a)) \leq \frac{\epsilon}{1 + \epsilon}, \tag{16}$$

*where $D_{\mathrm{TV}}$ is total variation distance, $d^P(s, a)$ is true stationary distribution of the preferred policy and $\hat{d}^P(s, a)$ is estimated stationary distribution which is defined by $\hat{\alpha}$ as $\hat{d}^P(s, a) = \frac{d^U(s,a) - \hat{\alpha} d^N(s,a)}{1 - \hat{\alpha}}$. (Proof in Appendix A.)*

*Proof.* Let, $\alpha^*$ is a true ratio between the stationary distribution of expert and anti-expert policy:

$$d^U(s, a) = (1 - \alpha^*)d^P(s, a) + \alpha^* d^N(s, a). \tag{18}$$

Then, $\hat{\alpha}$ can be represented by using $\alpha^*$ as follows:

$$\hat{\alpha} = \min_{(s,a)} \frac{d^U(s, a)}{d^N(s, a)} \tag{19}$$

$$= \min_{(s,a)} \frac{(1 - \alpha^*)d^P(s, a) + \alpha^* d^N(s, a)}{d^N(s, a)} \tag{20}$$

$$= \alpha^* + (1 - \alpha^*) \min_{(s,a)} \frac{d^P(s, a)}{d^N(s, a)}. \tag{21}$$

The estimated stationary distribution $\hat{d}^P$ with $\hat{\alpha}$ can be derived as follows:

$$\hat{d}^P(s,a) = \frac{d^U(s,a) - \hat{\alpha}d^N(s,a)}{1 - \hat{\alpha}} \tag{22}$$

$$= \frac{d^U(s,a) - (\alpha^* + (1-\alpha^*)\min_{(s,a)}\frac{d^P(s,a)}{d^N(s,a)})d^N(s,a)}{1 - \alpha^* + (1-\alpha^*)\min_{(s,a)}\frac{d^P(s,a)}{d^N(s,a)}} \tag{23}$$

$$= \frac{(1-\alpha^*)d^P(s,a) - (1-\alpha^*)\min_{(s,a)}\frac{d^P(s,a)}{d^N(s,a)}d^N(s,a)}{(1-\alpha^*) + (1-\alpha^*)\min_{(s,a)}\frac{d^P(s,a)}{d^N(s,a)}} \tag{24}$$

$$= \frac{d^P(s,a) - \min_{(s,a)}\frac{d^P(s,a)}{d^N(s,a)}d^N(s,a)}{1 + \min_{(s,a)}\frac{d^P(s,a)}{d^N(s,a)}} \tag{25}$$

$$= d^P(s,a) - \frac{\min_{(s,a)}\frac{d^P(s,a)}{d^N(s,a)}(d^P(s,a) + d^N(s,a))}{1 + \min_{(s,a)}\frac{d^P(s,a)}{d^N(s,a)}} \tag{26}$$

$$= d^P(s,a) - \frac{\min_{(s,a)}\frac{d^P(s,a)}{d^N(s,a)}(d^P(s,a) + d^N(s,a)) + (d^P(s,a) + d^N(s,a)) - (d^P(s,a) + d^N(s,a))}{1 + \min_{(s,a)}\frac{d^P(s,a)}{d^N(s,a)}} \tag{27}$$

$$= d^P(s,a) - \frac{(1 + \min_{(s,a)}\frac{d^P(s,a)}{d^N(s,a)})(d^P(s,a) + d^N(s,a)) - (d^P(s,a) + d^N(s,a))}{1 + \min_{(s,a)}\frac{d^P(s,a)}{d^N(s,a)}} \tag{28}$$

$$= -d^N(s,a) + \frac{d^P(s,a) + d^N(s,a)}{1 + \min_{(s,a)}\frac{d^P(s,a)}{d^N(s,a)}} \tag{29}$$

Finally, we can derive the total variation distance of the true and estimated stationary distribution of the expert policy:

$$D_{\mathrm{TV}}(d^P, \hat{d}^P) = \frac{1}{2}||d^P - \hat{d}^P||_1 \tag{30}$$

$$= \frac{1}{2}\sum_{(s,a)}\left|d^P(s,a) + d^N(s,a) - \frac{d^P(s,a) + d^N(s,a)}{1 + \min_{(s,a)}\frac{d^P(s,a)}{d^N(s,a)}}\right| \tag{31}$$

$$= \frac{1}{2}\sum_{(s,a)}\left(d^P(s,a) + d^N(s,a) - \frac{d^P(s,a) + d^N(s,a)}{1 + \min_{(s,a)}\frac{d^P(s,a)}{d^N(s,a)}}\right) \tag{32}$$

$$= \frac{1}{2}\left(\frac{\min_{(s,a)}\frac{d^P(s,a)}{d^N(s,a)}}{1 + \min_{(s,a)}\frac{d^P(s,a)}{d^N(s,a)}}\right)\sum_{(s,a)}(d^P(s,a) + d^N(s,a)) \tag{33}$$

$$= \frac{\min_{(s,a)}\frac{d^P(s,a)}{d^N(s,a)}}{1 + \min_{(s,a)}\frac{d^P(s,a)}{d^N(s,a)}} \tag{34}$$

$$\leq \frac{\epsilon}{1 + \epsilon}. \tag{35}$$

$\square$

## A.3 Proof that the reward function is well defined with selected $\alpha$

**Proposition A.1.** *If we select $\alpha$ as $\hat{\alpha} = \min_{(s,a)}\frac{d^U(s,a)}{d^N(s,a)}$, then the value inside the log function in Eq.* (10) *is always non-negative:*

$$\frac{1 - (1+\alpha)c^*(s,a)}{(1-\alpha)(1 - c^*(s,a))} \geq 0,$$

*where* $c^*(s,a) = \frac{d^N(s,a)}{d^U(s,a)+d^N(s,a)}$.

*Proof.* From the $\alpha$ selection rule, $\hat{\alpha} \leq \frac{d^U(s,a)}{d^N(s,a)}$ for all $(s,a)$. We can reformulate this inequation as $d^U(s,a) - \hat{\alpha}d^N(s,a) \geq 0$. Then the value inside the log function in Eq. (10) can be derived as follows:

$$\begin{aligned}
\frac{1-(1+\alpha)c^*(s,a)}{(1-\alpha)(1-c^*(s,a))} &= \frac{1-(1+\alpha)\frac{d^N(s,a)}{d^U(s,a)+d^N(s,a)}}{(1-\alpha)(1-\frac{d^N(s,a)}{d^U(s,a)+d^N(s,a)})} \\
&= \frac{1-(1+\alpha)\frac{d^N(s,a)}{d^U(s,a)+d^N(s,a)}}{(1-\alpha)\frac{d^U(s,a)}{d^U(s,a)+d^N(s,a)}} \\
&= \frac{d^U(s,a)+d^N(s,a)-(1+\alpha)d^N(s,a)}{(1-\alpha)d^U(s,a)} \\
&= \frac{d^U(s,a)-\alpha d^N(s,a)}{(1-\alpha)d^U(s,a)} \\
&= \frac{d^U(s,a)-\alpha d^N(s,a)}{(1-\alpha)d^U(s,a)} \\
&\geq 0
\end{aligned}$$

To avoid the case where this value is 0, in practical implementation, we select the $\alpha$ as $\hat{\alpha} = \min_{(s,a)} \frac{d^U(s,a)}{d^N(s,a)} - \epsilon$, where the $\epsilon$ is a small positive value ($\epsilon = 10^{-5}$). $\square$

## B    Pseudocode of SafeDICE

---

**Algorithm 1** SafeDICE

---

**Input:** A labeled non-preferred demonstrations $D^N$ , an unlabeled demonstrations $D^U$, a discriminator $c_\theta$ with parameter $\theta$, a neural network $\nu_\phi$ with parameter $\phi$, a policy network $\pi_\psi$ with parameter $\psi$

1: Pretrain the discriminator by optimizing the Eq. (9):
$$\theta^* = \arg\max_\theta \mathbb{E}_{d^N}[\log c_\theta(s,a)] + \mathbb{E}_{d^U}[\log(1 - c_\theta(s,a))]$$

2: Set the $\alpha$ as $\hat{\alpha}$ from Eq. (15) on $D^U$:
$$\hat{\alpha} = \min_{(s,a)\in D^U} \left( \frac{1}{c_{\theta^*}(s,a)} - 1 \right)$$

3: **for** each iteration $i$ **do**
4:     Update $\nu_\phi$ by optimizing the Eq. (13):
$$\min_\phi (1-\gamma)\mathbb{E}_{p_0}[\nu_\phi(s)] + \mathbb{E}_{d^U}\left[ \exp(A_{\nu_\phi}(s,a) - 1) \right]$$

5:     Update policy $\pi_\psi$ by weighted behavior cloning of unlabeled demonstrations $D^U$:
$$\min_\psi -\mathbb{E}_{(s,a)\sim D^U}[w^*(s,a)\log \pi_\psi(a|s)]$$

6: **end for**
**Output:** $\pi_\psi \approx \pi^*$

---

## C    Implementation Details of SafeDICE and Baseline Algorithms

In this section, we provide the implementation details of SafeDICE and baseline algorithms. Our code is available on `https://github.com/jys5609/SafeDICE`.

### C.1    SafeDICE

We implement SafeDICE based on the Real-World RL (RWRL) suite [8] and codebase of DemoDICE [13], which is one of recent DICE-based offline IL algorithms. Similar to DemoDICE [13], due to the instability issue of Eq. (13), we implemented an alternative objective that is more numerically stable while having the same optimal solution:

$$\min_\nu \mathcal{L}(w_\nu^*, \nu; r) = (1-\gamma)\mathbb{E}_{p_0}[\nu(s)] + \log\mathbb{E}_{d^U}\left[ \exp\left(A_\nu(s,a)\right) \right]. \tag{36}$$

Then, we estimate Eq. (14) using self-normalized importance sampling [23] as follows:

$$\min_\pi -\frac{\mathbb{E}_{(s,a)\sim D^U}[\tilde{w}_{\tilde{\nu}^*}(s,a)\log \pi_\psi(a|s)]}{\mathbb{E}_{(s,a)\sim D^U}[\tilde{w}_{\tilde{\nu}^*}(s,a)]}, \tag{37}$$

where $\tilde{\nu}^*$ is optimal solution of Eq. (36) and $\tilde{w}_{\tilde{\nu}^*} := \exp(A_{\tilde{\nu}^*}(s,a))$.

### C.2    Discriminator-Weighted Behavior Cloning (DWBC)

We modified DWBC, which adopted positive-unlabeled learning, to be considered as a baseline algorithm in our experimental settings. Similar to positive-unlabeled learning, we adopted negative-unlabeled learning to the objective of discriminator learning as follows:

$$\min_\theta \ \eta \ \mathbb{E}_{(s,a)\sim D^N}[-\log d_\theta(s,a,\log \pi)]$$
$$+\mathbb{E}_{(s,a)\sim D^U}[-\log(1 - d_\theta(s,a,\log \pi))]$$
$$-\eta \ \mathbb{E}_{(s,a)\sim D^N}[-\log(1 - d_\theta(s,a,\log \pi))], \tag{38}$$

where $d_\theta$ is a discriminator with parameter $\theta$, and $\eta$ is a hyperparameter that corresponds to the proportion of negative samples to unlabeled samples.

Based on the trained discriminator by Eq. (38), we adopted weighted behavior cloning on the unlabeled demonstrations for the policy learning as follows:

$$\min_\psi \; -\mathbb{E}_{(s,a)\sim D^U}[(1 - d_\theta(s,a)) \log \pi_\psi(a|s)]. \tag{39}$$

### C.3 Preference-based Policy Learning (PPL)

We trained the reward function with the following loss function follows from the Bradley-Terry model [3] which is commonly used in preference-based policy learning approaches [4, 18]:

$$\min_\theta \; -\sum_{(i,j)\in\mathcal{P}} \log \frac{\exp\left(\sum_{(s,a)\in\tau_i} R_\theta(s,a)\right)}{\exp\left(\sum_{(s,a)\in\tau_i} R_\theta(s,a)\right) + \exp\left(\sum_{(s,a)\in\tau_j} R_\theta(s,a)\right)}, \tag{40}$$

where $\tau_i$ and $\tau_j$ represent trajectories, and $\mathcal{P} = \{(i,j) : \tau_i \succ \tau_j\}$. In our problem setting, we adopt this model assuming that unlabeled demonstrations are better than anti-expert demonstrations (i.e. $\tau_i \succ \tau_j$ for all $\tau_i \in D^U$ and $\tau_j \in D^N$).

Based on the trained reward model by Eq. (40), we adopt weighted behavior cloning on the unlabeled demonstrations for the policy learning as follows:

$$\min_\psi \; -\mathbb{E}_{(s,a)\sim D^U}[R_\theta(s,a) \log \pi_\psi(a|s)]. \tag{41}$$

### C.4 DExperts

**Tabular MDPs** For the tabular MDPs, we separately trained unlabeled policy $\pi^U$ and non-preferred policy $\pi^N$ with a behavior cloning model using unlabeled demonstrations and non-preferred demonstrations as follows:

$$\min_{\pi^U} \; -\mathbb{E}_{(s,a)\sim D^U}[\log \pi^U(a|s)], \tag{42}$$

$$\min_{\pi^N} \; -\mathbb{E}_{(s,a)\sim D^N}[\log \pi^N(a|s)]. \tag{43}$$

Then, as the final policy, the ensemble of $\pi^U$ and $\pi^N$ at decoding-time is used:

$$\pi(a|s) = \pi^U(a|s) - \alpha\pi^N(a|s), \tag{44}$$

where $\alpha$ is a hyperparameter that controls the degree of leveraging the non-preferred policy.

**Continuous MDPs** For the continuous MDPs, it is not straightforward to apply DExperts as a decoding-time method. Therefore, we implemented DExperts with the following objective that is similarly considered at training time:

$$\min_\psi \; -\mathbb{E}_{(s,a)\sim D^U}[\log \pi_\psi(a|s)] + \alpha\,\mathbb{E}_{(s,a)\sim D^N}[\log \pi_\psi(a|s)], \tag{45}$$

where $\alpha$ is a hyperparameter that controls the degree of leveraging the non-preferred demonstrations.

# D Experimental Details

## D.1 Experimental details for RWRL suite

**Task specification** We conduct experiments on the RWRL suite with safety constraints. Table 1 shows the tasks, safety constraint settings, and the information of dataset for each domain that we used in our experiments.

| Task specification | Cartpole | Walker | Quadruped |
|---|---|---|---|
| name of the used task | realworld_swingup | realworld_walk | realworld_walk |
| name of the used safety constraint | slider_pos_constraint | joint_velocity_constraint | joint_velocity_constraint |
| safety coefficient | 0.2 | 0.2 | 0.35 |
| # of unlabeled demonstrations | 5000 | 2000 | 2000 |
| # of labeled non-preferred demonstrations | 80 | 10 | 50 |
| mean cost of preferred demonstrations | 71.95 | 56.14 | 97.01 |
| mean cost of non-preferred demonstrations | 316.75 | 377.58 | 323.60 |
| mean return of preferred demonstrations | 516.28 | 939.74 | 991.11 |
| mean return of non-preferred demonstrations | 519.82 | 940.06 | 989.59 |

Table 1: Task specification of each domain used in our experimental results on RWRL environment.

**Hyperparameter configurations** For a fair comparison, we use the same architecture and learning rate to train the policy networks, discriminators, and reward models of each algorithm. Table 2 summarizes the hyperparameter settings that we used in our experiments. The additional hyperparameters $\eta$ and $\alpha$ of DWBC and DExperts were selected by the best performance among $\{0.01, 0.02, 0.05, 0.1, 0.2, 0.4, 0.6, 0.8\}$.

| Hyperparameters | BC | DWBC | PPL | DExperts | SafeDICE |
|---|---|---|---|---|---|
| $\gamma$ (discount factor) | 0.99 | 0.99 | 0.99 | 0.99 | 0.99 |
| learning rate (actor) | $3 \times 10^{-4}$ | $3 \times 10^{-4}$ | $3 \times 10^{-4}$ | $3 \times 10^{-4}$ | $3 \times 10^{-4}$ |
| network size (actor) | [256, 256] | [256, 256] | [256, 256] | [256, 256] | [256, 256] |
| learning rate (critic) | - | - | - | - | $3 \times 10^{-4}$ |
| network size (critic) | - | - | - | - | [256, 256] |
| learning rate (discriminator / reward model) | - | $3 \times 10^{-4}$ | $3 \times 10^{-4}$ | - | $3 \times 10^{-4}$ |
| network size (discriminator / reward model) | - | [256,256] | [256,256] | - | [256, 256] |
| gradient penalty coefficient (discriminator) | - | 0.1 | - | - | 0.1 |
| batch size | 512 | 512 | 512 | 512 | 512 |
| # of training iterations | 1,200,000 | 1,200,000 | 1,200,000 | 1,200,000 | 1,200,000 |

Table 2: Configurations of hyperparameters used in our experimental results on RWRL environment.

### D.2 Experimental details for Safety Gym

**Task specification** We conduct experiments on the Safety Gym with safety constraints. Table 3 shows the tasks, safety constraint settings, and the information of dataset for each domain that we used in our experiments.

| Task specification | Goal | Button |
|---|---|---|
| type of robot | Point | Point |
| level of difficulty | 1 | 1 |
| # of unlabeled demonstrations | 2000 | 2000 |
| # of labeled non-preferred demonstrations | 50 | 50 |
| mean cost of preferred demonstrations | 13.798 | 12.085 |
| mean cost of non-preferred demonstrations | 107.977 | 166.099 |
| mean return of preferred demonstrations | 19.911 | 8.286 |
| mean return of non-preferred demonstrations | 20.018 | 21.933 |

Table 3: Task specification of each domain used in our experimental results on Safety Gym environment.

**Hyperparameter configurations** For a fair comparison, we use the same architecture and learning rate to train the policy networks, discriminators, and reward models of each algorithm. Table 4 summarizes the hyperparameter settings that we used in our experiments. The additional hyperparameters $\eta$ and $\alpha$ of DWBC and DExperts were selected by the best performance among $\{0.01, 0.02, 0.05, 0.1, 0.2, 0.4, 0.6, 0.8\}$.

| Hyperparameters | BC | DWBC | PPL | DExperts | SafeDICE |
|---|---|---|---|---|---|
| $\gamma$ (discount factor) | 0.99 | 0.99 | 0.99 | 0.99 | 0.99 |
| learning rate (actor) | $3 \times 10^{-4}$ | $3 \times 10^{-4}$ | $3 \times 10^{-4}$ | $3 \times 10^{-4}$ | $3 \times 10^{-4}$ |
| network size (actor) | [256, 256] | [256, 256] | [256, 256] | [256, 256] | [256, 256] |
| learning rate (critic) | - | - | - | - | $3 \times 10^{-4}$ |
| network size (critic) | - | - | - | - | [256, 256] |
| learning rate (discriminator / reward model) | - | $3 \times 10^{-4}$ | $3 \times 10^{-4}$ | - | $3 \times 10^{-4}$ |
| network size (discriminator / reward model) | - | [256,256] | [256,256] | - | [256, 256] |
| gradient penalty coefficient (discriminator) | - | 0.1 | - | - | 0.1 |
| batch size | 512 | 512 | 512 | 512 | 512 |
| # of training iterations | 1,000,000 | 1,000,000 | 1,000,000 | 1,000,000 | 1,000,000 |

Table 4: Configurations of hyperparameters used in our experimental results on Safety Gym environment.

# E    Additional Results on RWRL suite

## E.1    Results of CVaR $k\%$ Cost on RWRL suite

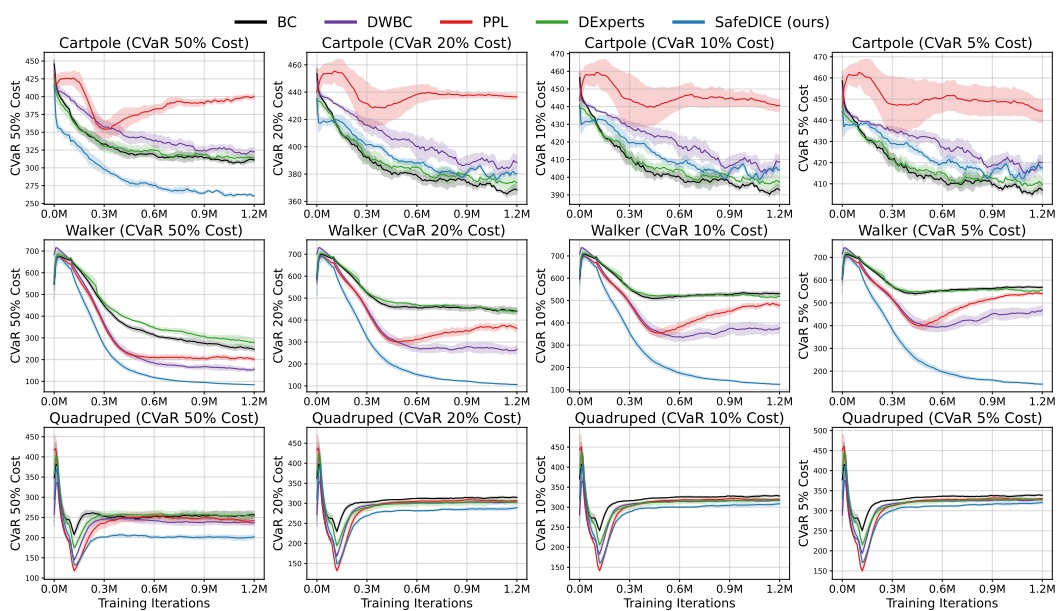

Figure 5: Additional experimental results of Figure 2 on metrics of CVaR $k\%$ Cost .

### E.2  Results of DWBC and DExperts with various hyperparameters on RWRL Walker domain

To study the hyperparameter sensitivity of DWBC and DExperts, we demonstrate the results of DWBC and DExperts with various hyperparameter settings. For DWBC, we compare the performance of DWBC with different values of $\eta \in \{0.01, 0.02, 0.05, 0.1, 0.2, 0.4, 0.6, 0.8\}$ and DWBC without $\eta$ (i.e. naive discriminator training without negative-unlabeled learning). As shown in Figure 6, DWBC shows very sensitive performance to hyperparameter $\eta$ and unstable learning that sometimes even suffers from degeneration issues.

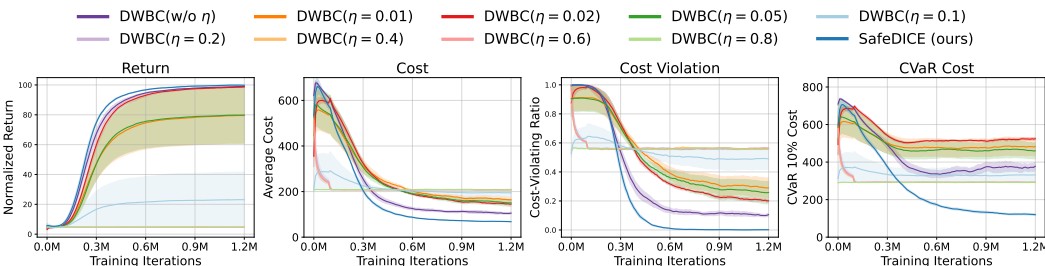

Figure 6: Additional experimental results of DWBC with various hyperparameters on Walker domain.

We also compare the performance of DExperts with different values of $\alpha \in \{0, 0.01, 0.02, 0.05, 0.1, 0.2, 0.4, 0.6, 0.8\}$. Here, DExperts($\alpha = 0$) is reduced to the naive behavior cloning model on unlabeled demonstrations. As shown in Figure 7, DExperts shows very sensitive performance to hyperparameter $\alpha$ and is vulnerable to degeneration issues.

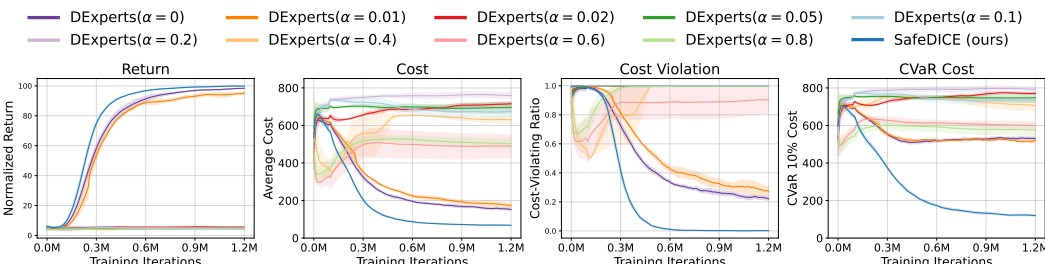

Figure 7: Additional experimental results of DExperts with various hyperparameters on Walker domain.

## E.3 Results with varying amount of labeled non-preferred demonstrations

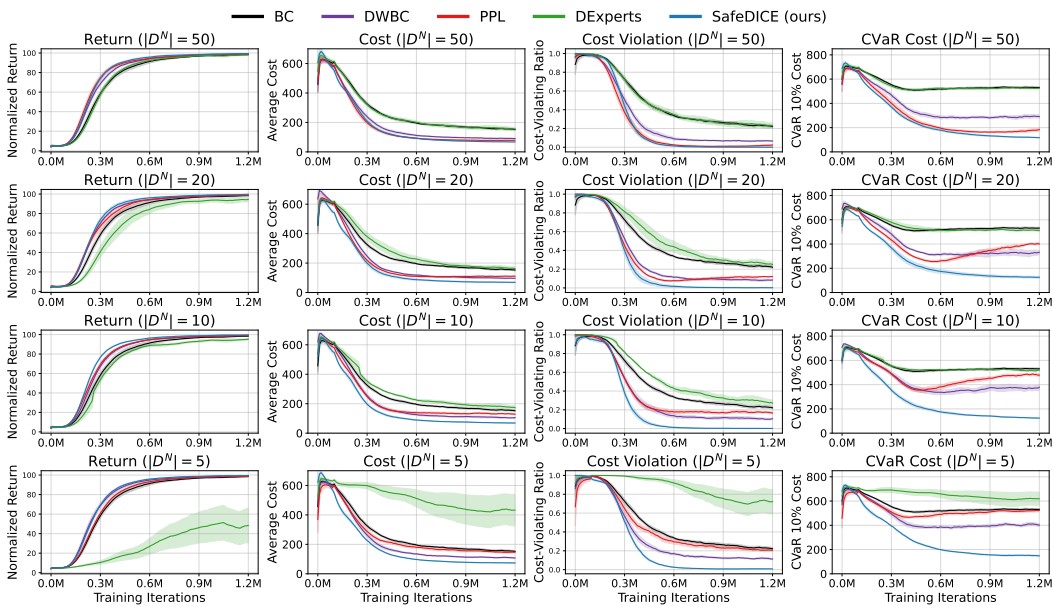

Figure 8: Experimental results on RWRL Walker domain with different numbers of labeled non-preferred demonstrations ($|D^N| \in \{50, 20, 10, 5\}$). All results are averaged over 5 runs, and the shaded area represents the standard error.

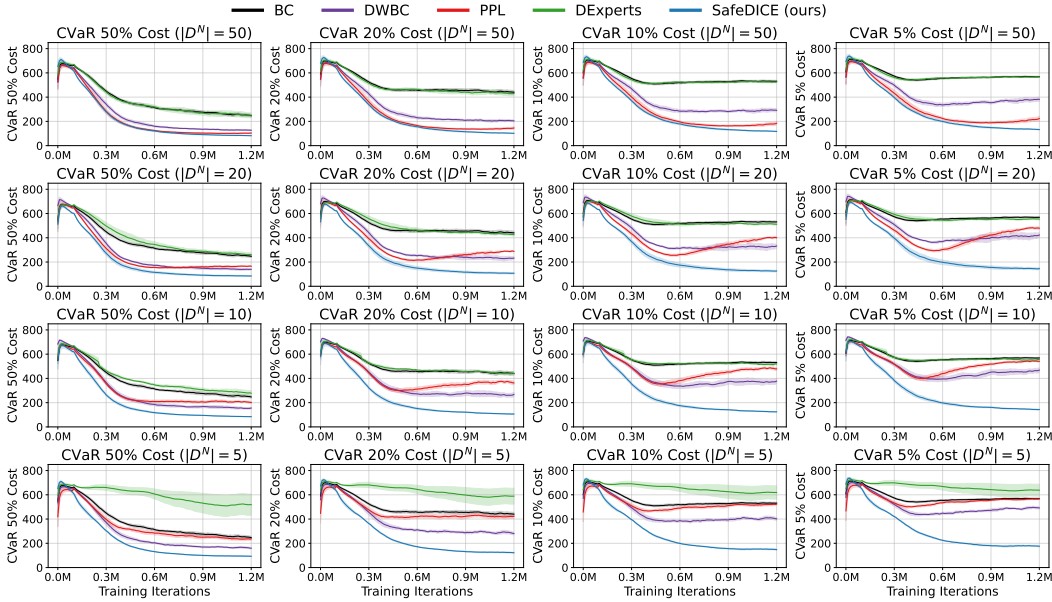

Figure 9: Additional experimental results of Figure 8 on metrics of CVaR $k\%$ Cost.

## E.4 Results of SafeDICE with various $\alpha$

To show the importance of theoretically determining $\alpha$, we show additional results of SafeDICE with various values of $\alpha$. According to the value of $\alpha$, the reward function $r_\alpha(s, a)$ defined by Eq. (10) cannot guarantee that the value inside the log function always becomes a positive. Therefore, we simply performed learning with a reward function that artificially clipped the value inside the log function of Eq. (10). As shown in Figure 10 and 11, the results show various performance depending on the value of $\alpha$, and result with $\alpha$ value selected by Eq. (15) shows the best performance.

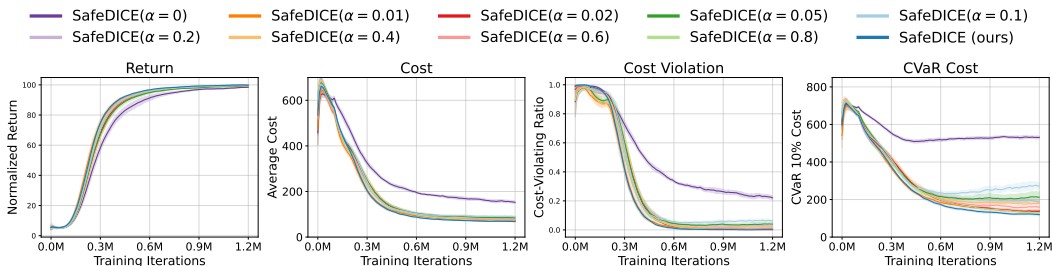

Figure 10: Additional experimental results of SafeDICE with various hyperparameters on Walker domain.

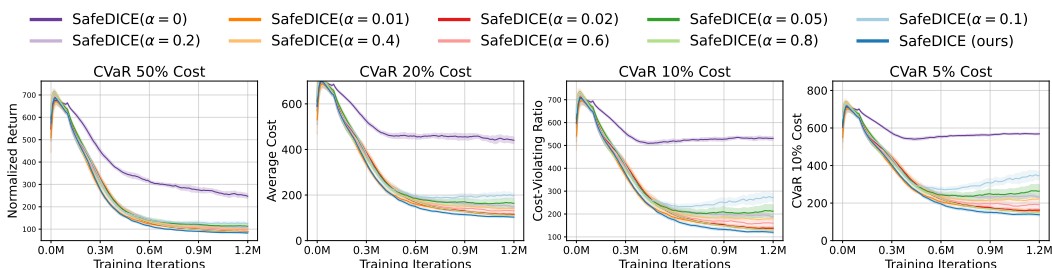

Figure 11: Additional experimental results of SafeDICE with various hyperparameters on Walker domain.

## E.5 Impact of the stationary distribution constraint and labeled non-preferred demonstrations

We conducted additional experiments to evaluate the impact of the stationary distribution constraint and labeled non-preferred demonstrations. As can be seen from the Figure 12, the performance is degraded both in the absence of stationary distribution constraints and in the case of not using labeled non-preferred demonstrations.

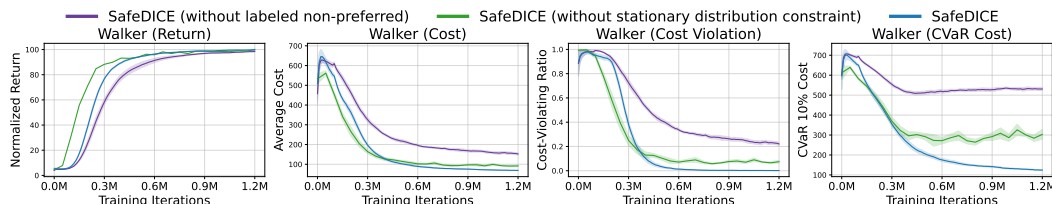

Figure 12: Experimental results on RWRL Walker domain without stationary distribution constraint and labeled non-preferred demonstrations. All results are averaged over 5 independent runs, and the shaded area represents the standard error.

**E.6 Results with various configurations of unlabeled demonstrations**

In order to evaluate our algorithm with more various configurations of unlabeled demonstrations, we also conducted additional experiments for various configurations of unlabeled demonstrations consisting of different ratios of preferred and non-preferred demonstrations $((D_P^U, D_N^U) \in \{(250, 1000), (500, 1000), (1000, 500), (1000, 250)\}$, where $|D_P^U|$ and $|D_N^U|$ denote the number of preferred and non-preferred demonstrations in $D^U$, respectively). As can be seen from the Figure 13, the results show that SafeDICE outperforms baselines in all settings with various configurations of unlabeled demonstrations. These results show that SafeDICE can be applied more effectively in various real-world scenarios regardless of the preferred and non-preferred ratio of unlabeled demonstrations.

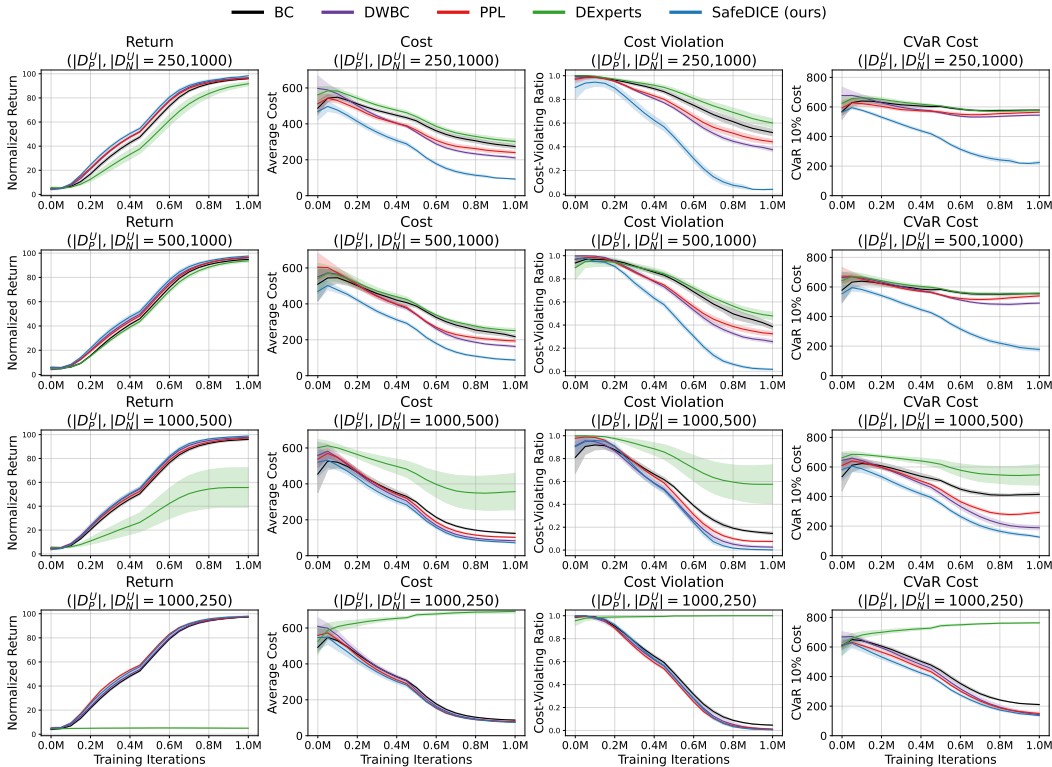

Figure 13: Experimental results on RWRL Walker domain with different numbers of preferred and non-preferred demonstrations in unlabeled demonstrations $D^U$. $|D_P^U|$ and $|D_N^U|$ denote the number of preferred and non-preferred demonstrations in $D^U$ $((D_P^U, D_N^U) \in \{(250, 1000), (500, 1000), (1000, 500), (1000, 250)\})$. All results are averaged over 5 independent runs, and the shaded area represents the standard error.

# F   Comparison with constrained RL algorithm

To compare with the safe RL algorithm as an *oracle* agent, we conducted additional experiments for an offline-constrained RL method. We run COptiDICE [16], the state-of-the-art offline constrained RL algorithm, using the datasets that are augmented with additional (ground-truth) reward and cost annotations. Table 5-9 summarize overall performance of SafeDICE and baseline algorithms including COptiDICE. The results show that SafeDICE is competitive with (or even outperforms in some domains) the constrained offline RL algorithm COptiDICE, even though SafeDICE uses only a much smaller amount of annotation information.

## F.1   Experimental results on RWRL control tasks

| CARTPOLE | Normalized Return | Average Cost | Cost Violation | CVaR 10% Cost |
|---|---|---|---|---|
| BC | $100.24 \pm 0.40$ | $184.44 \pm 0.94$ | $0.48 \pm 0.00$ | $392.75 \pm 2.72$ |
| DWBC | $99.78 \pm 0.55$ | $189.97 \pm 3.21$ | $0.47 \pm 0.00$ | $408.71 \pm 4.01$ |
| PPL | $83.70 \pm 0.87$ | $229.80 \pm 1.64$ | $0.48 \pm 0.01$ | $440.68 \pm 3.49$ |
| DExperts | $100.28 \pm 0.38$ | $186.15 \pm 2.29$ | $0.48 \pm 0.01$ | $397.05 \pm 3.11$ |
| SafeDICE | $99.91 \pm 0.60$ | $154.88 \pm 1.79$ | $0.34 \pm 0.00$ | $404.22 \pm 1.56$ |
| COptiDICE | $\mathbf{107.95 \pm 2.10}$ | $\mathbf{97.59 \pm 7.21}$ | $\mathbf{0.06 \pm 0.01}$ | $\mathbf{183.63 \pm 6.18}$ |

Table 5: Experimental results on Cartpole domain. All results indicate averages and standard errors over 5 independent runs.

| WALKER | Normalized Return | Average Cost | Cost Violation | CVaR 10% Cost |
|---|---|---|---|---|
| BC | $98.41 \pm 0.11$ | $152.25 \pm 5.33$ | $0.22 \pm 0.01$ | $531.07 \pm 7.53$ |
| DWBC | $98.99 \pm 0.10$ | $105.63 \pm 3.85$ | $0.11 \pm 0.01$ | $378.01 \pm 20.88$ |
| PPL | $99.02 \pm 0.10$ | $128.46 \pm 4.84$ | $0.17 \pm 0.01$ | $478.75 \pm 12.53$ |
| DExperts | $95.21 \pm 0.75$ | $173.20 \pm 11.22$ | $0.27 \pm 0.03$ | $518.38 \pm 5.25$ |
| SafeDICE | $\mathbf{99.67 \pm 0.13}$ | $\mathbf{68.90 \pm 0.83}$ | $\mathbf{0.00 \pm 0.00}$ | $\mathbf{124.54 \pm 2.56}$ |
| COptiDICE | $98.07 \pm 0.34$ | $77.86 \pm 2.14$ | $0.01 \pm 0.00$ | $159.91 \pm 8.90$ |

Table 6: Experimental results on Walker domain. All results indicate averages and standard errors over 5 independent runs.

| QUADRUPED | Normalized Return | Average Cost | Cost Violation | CVaR 10% Cost |
|---|---|---|---|---|
| BC | $\mathbf{100.19 \pm 0.08}$ | $178.10 \pm 5.20$ | $0.38 \pm 0.02$ | $328.10 \pm 1.83$ |
| DWBC | $99.87 \pm 0.16$ | $167.15 \pm 3.32$ | $0.34 \pm 0.02$ | $316.86 \pm 1.74$ |
| PPL | $99.87 \pm 0.09$ | $169.93 \pm 5.06$ | $0.35 \pm 0.03$ | $319.77 \pm 1.70$ |
| DExperts | $97.05 \pm 2.62$ | $178.75 \pm 4.78$ | $0.38 \pm 0.03$ | $317.55 \pm 4.40$ |
| SafeDICE | $99.68 \pm 0.25$ | $148.21 \pm 2.84$ | $0.24 \pm 0.01$ | $308.71 \pm 2.90$ |
| COptiDICE | $87.37 \pm 3.19$ | $\mathbf{130.21 \pm 5.20}$ | $\mathbf{0.16 \pm 0.03}$ | $\mathbf{263.41 \pm 17.00}$ |

Table 7: Experimental results on Quadruped domain. All results indicate averages and standard errors over 5 independent runs.

### F.2 Experimental results on Safety Gym control tasks

| GOAL | Normalized Return | Average Cost | Cost Violation | CVaR 10% Cost |
|---|---|---|---|---|
| BC | $94.08 \pm 0.60$ | $51.47 \pm 1.49$ | $0.38 \pm 0.02$ | $109.66 \pm 1.63$ |
| DWBC | $93.93 \pm 0.53$ | $47.96 \pm 1.77$ | $0.31 \pm 0.02$ | $108.79 \pm 2.12$ |
| PPL | $94.43 \pm 0.66$ | $52.31 \pm 1.52$ | $0.37 \pm 0.02$ | $111.41 \pm 2.37$ |
| DExperts | $87.76 \pm 4.41$ | $50.82 \pm 1.55$ | $0.36 \pm 0.02$ | $112.62 \pm 1.54$ |
| SafeDICE | $92.04 \pm 0.44$ | $\mathbf{39.49 \pm 0.94}$ | $\mathbf{0.21 \pm 0.02}$ | $\mathbf{93.35 \pm 1.45}$ |
| COptiDICE | $92.98 \pm 1.13$ | $48.79 \pm 0.80$ | $0.32 \pm 0.01$ | $107.25 \pm 0.78$ |

Table 8: Experimental results on Goal task. All results indicate averages and standard errors over 5 independent runs.

| BUTTON | Normalized Return | Average Cost | Cost Violation | CVaR 10% Cost |
|---|---|---|---|---|
| BC | $88.64 \pm 1.08$ | $98.18 \pm 2.19$ | $0.49 \pm 0.01$ | $220.87 \pm 3.48$ |
| DWBC | $85.53 \pm 0.80$ | $96.31 \pm 2.38$ | $0.47 \pm 0.02$ | $225.49 \pm 3.52$ |
| PPL | $86.76 \pm 1.06$ | $99.97 \pm 2.19$ | $0.50 \pm 0.02$ | $229.46 \pm 3.30$ |
| DExperts | $86.28 \pm 1.35$ | $97.30 \pm 1.70$ | $0.49 \pm 0.01$ | $225.93 \pm 5.34$ |
| SafeDICE | $71.48 \pm 1.33$ | $\mathbf{70.67 \pm 1.61}$ | $\mathbf{0.30 \pm 0.01}$ | $\mathbf{185.64 \pm 1.54}$ |
| COptiDICE | $\mathbf{94.09 \pm 1.68}$ | $111.10 \pm 3.53$ | $0.58 \pm 0.02$ | $238.46 \pm 3.44$ |

Table 9: Experimental results on Button task. All results indicate averages and standard errors over 5 independent runs.

# G Tabular MDPs Results

In order to show the motivation of the paper and the limitation of existing algorithms through illustrative examples, we conducted the experiment in tabular MDPs, which extends the Four Rooms environment [28, 22].

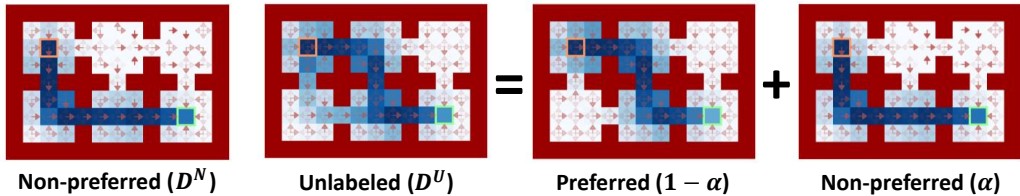

**Non-preferred ($D^N$)**      **Unlabeled ($D^U$)**      **Preferred ($1 - \alpha$)**      **Non-preferred ($\alpha$)**

Figure 14: Illustrative examples of experimental settings on Six Rooms environment. We assume the pre-collected dataset which consists of scarce but labeled non-preferred demonstrations $D^N$, and abundant but unlabeled demonstrations $D^U$ from both preferred and non-preferred policies.

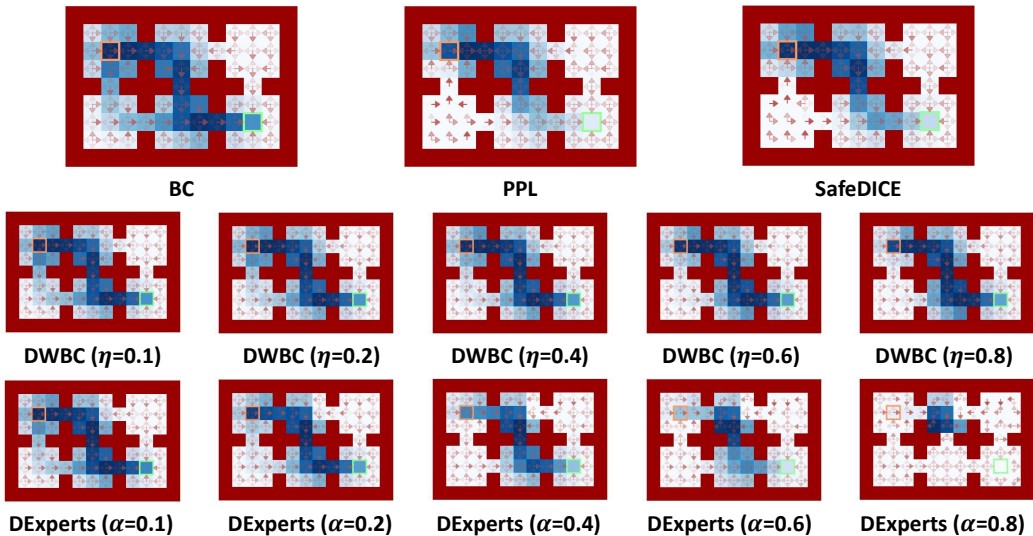

Figure 15: Experimental results on Six Room environment. We demonstrate the qualitative results of each algorithm learned using the dataset given in Figure 14. For DWBC and DExperts that require additional hyperparameter search, we show the results with various hyperparameter settings. $\eta$ and $\alpha$ represent hyperparameters that control the degree of leveraging the non-preferred demonstrations of DWBC and DExperts, respectively.

**Task setup** We consider the Six Rooms environment, a variant of the Four Rooms environment [28, 22], in which an agent aims to navigate to a goal location within a gridworld. Figure 14 demonstrates our experimental settings on the Six Rooms environment. The orange and green squares denote the initial state and goal state, respectively. The arrows for each action at the square represent the policy, where the opacity of the arrow is determined by the probability $\pi(a|s)$. The opacity of each square represents the state marginals of each stationary distribution. In order to compose the task naturally, we assume that the preferred and non-preferred policies have both similar and different behaviors in some state-action spaces. We define the policy that reaches the goal state from the initial state through the downward direction as an non-preferred policy, which represents constraint-violating behavior, and the policy that reaches the goal state through the right direction as an preferred policy. In this task, the agent aims to learn the policy that recovers the preferred policy from unlabeled demonstrations while avoiding the constraint-violating behavior using labeled non-preferred demonstrations. Figure 14 shows a given dataset to the agent for learning, which consists of *scarce but labeled* non-preferred demonstrations $D^N$ and *abundant but unlabeled* demonstrations

$D^U$ from both preferred and non-preferred policy. The details for the experimental settings can be found in Appendix D.

**Results** Figure 15 summarizes the qualitative results of SafeDICE and baseline algorithms on Six Rooms environment. Naive BC on unlabeled demonstrations follows both preferred and non-preferred behavior since it does not consider avoiding the non-preferred demonstrations. On the other hand, PPL and SafeDICE show safe behaviors in which non-preferred demonstrations are completely avoided while following preferred demonstrations. For DWBC and DExperts that require additional hyperparameter search, we show the behaviors of learned policies with different hyperparameter settings. Among various hyperparameter settings, there are cases where an agent learns a policy close to an preferred behavior at a specific hyperparameter. However, the results show that both DWBC and DExperts are hyperparameter sensitive, which is unsuitable for real-world tasks where the deployment is costly or risky. In addition, DExperts not only have cost-violating behaviors but also cause *degeneration* issues that fall into abnormal behaviors that are completely different from given demonstrations. To show that SafeDICE robustly works in various problem settings, we also provide additional results including cases where the distributions of labeled and unlabeled non-preferred demonstrations are dissimilar (see Appendix G.1), and more complex environments (see Appendix G.2). As shown in Appendix G.1, SafeDICE learns a safe policy in which non-preferred behaviors are completely avoided while following preferred behavior, even when the distributions of labeled and unlabeled non-preferred demonstrations are dissimilar.

### G.1 Experimental results with the dissimilar distribution of labeled and unlabeled non-preferred demonstrations

To show that SafeDICE works robustly, we conduct the experiment considering a problem setting more similar to real-world tasks, where the distributions of unlabeled and labeled non-preferred demonstrations are dissimilar. In this experimental setting, we define the non-preferred policy is a multimodal policy that consists of a mixture of two different policies $\pi^{A_1}$ and $\pi^{A_2}$. We assume that non-preferred demonstrations of labeled and unlabeled are sampled from policies $\pi^{A_1}$ and $\pi^{A_2}$ with different ratios. As shown in Figure 16, baseline algorithms fail to completely recover the preferred behavior or follow the non-preferred behavior. On the other hand, SafeDICE shows safe behavior in which non-preferred behaviors are completely avoided while following preferred behavior, even when the distributions of labeled and unlabeled non-preferred demonstrations are dissimilar.

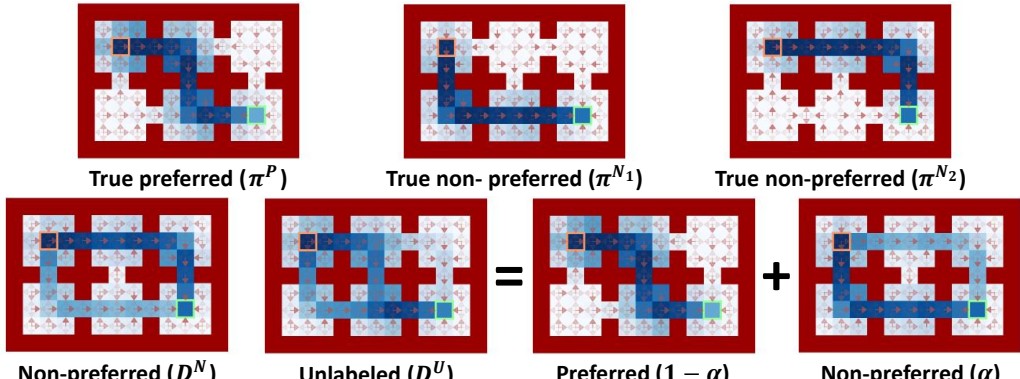

Figure 16: Illustrative examples of additional experimental settings on the tabular environment. We assume the pre-collected dataset which consists of scarce but labeled non-preferred demonstrations $D^N$ and abundant but unlabeled demonstrations $D^U$ from both preferred and non-preferred policies. In this experiment, we consider a more challenging problem setting where the distributions of labeled and unlabeled non-preferred demonstrations are dissimilar.

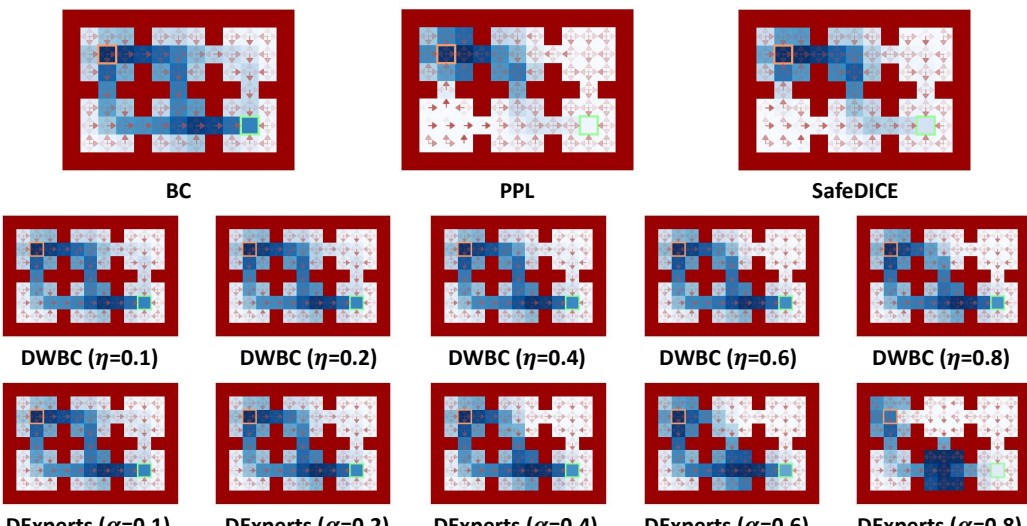

Figure 17: Experimental results on the tabular environment. We demonstrate the qualitative results of each algorithm learned using the dataset given in Figure 16.

## G.2 Additional results for various Tabular MDPs

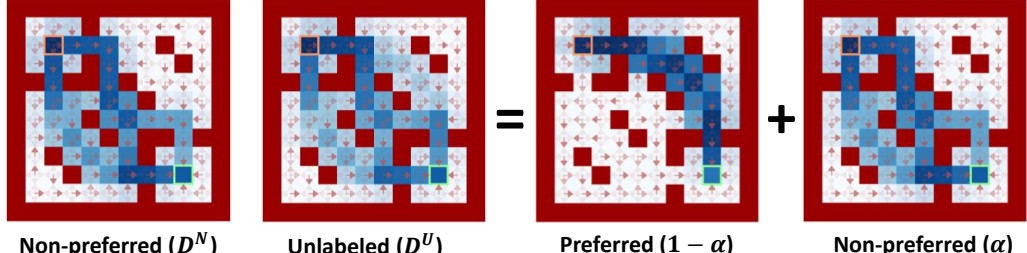

Figure 18: Illustrative examples of additional experimental settings on the tabular environment. We assume the pre-collected dataset which consists of scarce but labeled non-preferred demonstrations $D^N$ and abundant but unlabeled demonstrations $D^U$ from both preferred and non-preferred policies.

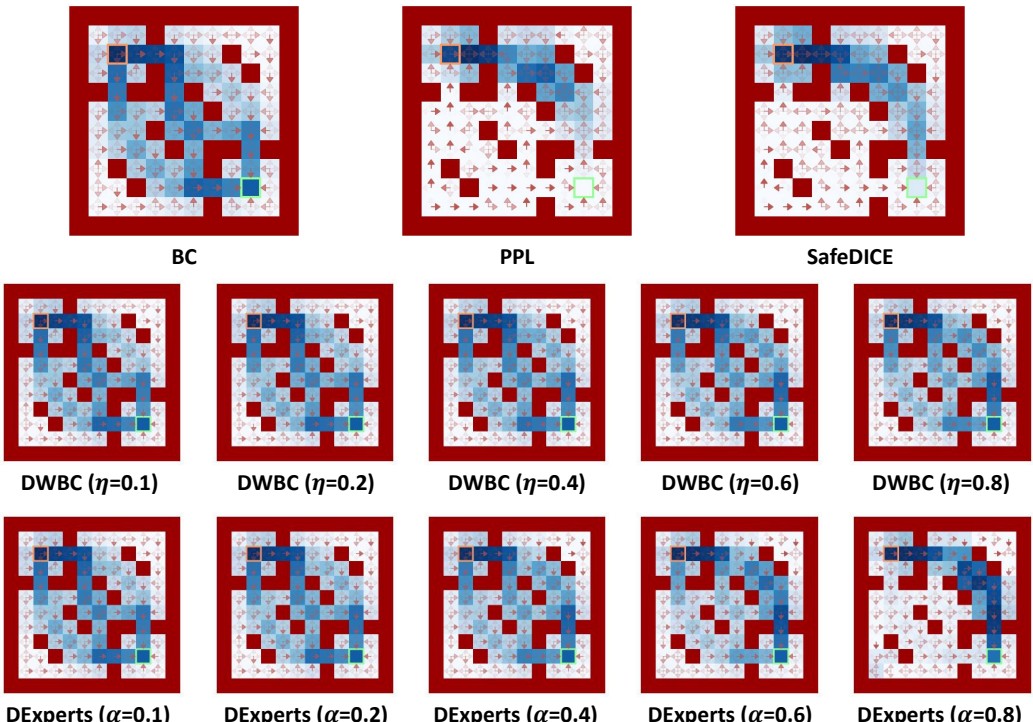

Figure 19: Experimental results on the tabular environment. We demonstrate the qualitative results of each algorithm learned using the dataset given in Figure 18. For DWBC and DExperts that require additional hyperparameter search, we show the results with various hyperparameter settings. $\eta$ and $\alpha$ represent hyperparameters that control the degree of leveraging the non-preferred demonstrations of DWBC and DExperts, respectively.

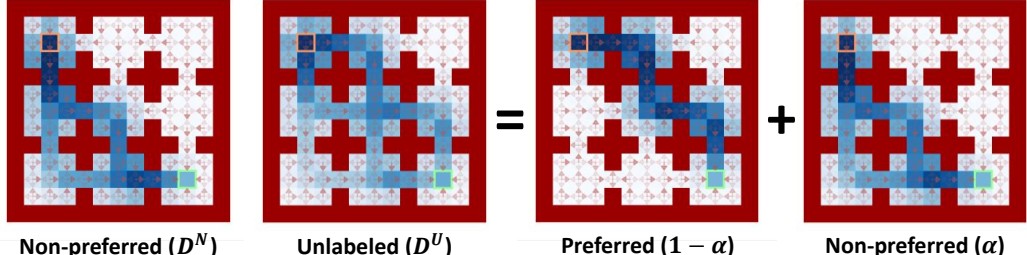

Figure 20: Illustrative examples of additional experimental settings on the tabular environment. We assume the pre-collected dataset which consists of scarce but labeled non-preferred demonstrations $D^N$ and abundant but unlabeled demonstrations $D^U$ from both preferred and non-preferred policies.

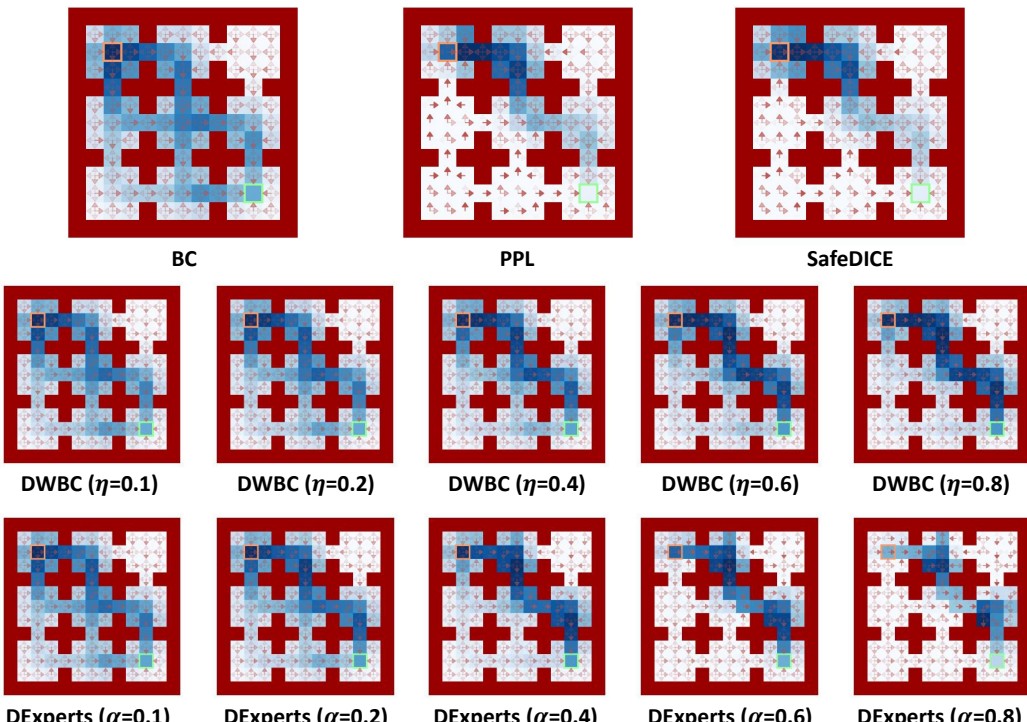

Figure 21: Experimental results on the tabular environment. We demonstrate the qualitative results of each algorithm learned using the dataset given in Figure 20. For DWBC and DExperts that require additional hyperparameter search, we show the results with various hyperparameter settings. $\eta$ and $\alpha$ represent hyperparameters that control the degree of leveraging the non-preferred demonstrations of DWBC and DExperts, respectively.

