# OpenReview forum: "SafeDICE: Offline Safe Imitation Learning with Non-Preferred Demonstrations"
_NeurIPS.cc/2023/Conference — NeurIPS 2023 poster_

### Official Review · Reviewer_Ajjv · 2023-06-17

**Soundness:** 3 good
**Presentation:** 3 good
**Contribution:** 2 fair
**Rating:** 6
**Confidence:** 4

**Summary:**

This paper presents an offline safe imitation learning algorithm called SafeDICE. A unique point of SafeDICE is that a safe policy is learned by non-preferred and unlabeled demonstrations in the imitation learning framework. Based on the formulations studied in DICE family, this paper formulated the safe IL problem as a stationary distribution matching problem, and then solve a single convex minimization problem without additional hyper-parameter search. In the experiments using RWRL and Safety-Gym benchmarks, the authors demonstrate the effectiveness of the SafeDICE algorithms and show that a better policy can be learned.

**Strengths:**

- This paper is well-written and easy to understand. I think the presentation of the ideas is clear and concise.

- The problem setting is well-motivated with a timely example (i.e., chatbot in lines 30-32). I agree that the problem setting is actually important and unexplored.

- The empirical evaluation of this paper is great. I think the empirical evaluation is sufficient because the authors demonstrated the effectiveness of the proposed SafeDICE algorithm in two environments with comparison with reasonable baselines.

**Weaknesses:**

- It is a little bit unclear what is the true mathematical contributions. As the authors describes as the following, but it seems that the most of the mathematical formulations are based on the DICE family papers and the contributions of this paper is a little bit minor.
> The main flow of our derivation follows prior DICE-based offline imitation learning methods [12, 13], but it is clearly different from the existing methods that require hyper-parameter search to deal with unknown $\alpha$.

- Though the authors demonstrated that the SafeDICE performs better than other baselines, the performance itself seems low if I understand correctly. And, since return is normalized, it is unclear even whether or not the tasks are solved. I think it would be better to add a typical Safe RL algorithm (e.g., CPO, TRPO-Lagrangian) as an oracle agent. Note that, I do *not* require the SafeDICE to perform better than Safe RL algorithms.

**Questions:**

[Q1] What is the actual value of the return without normalization? Did the authors confirm that the tasks are successfully solved?

[Q2] What is the threshold of the Safety Gym? Is it a default parameter (i.e., 20)?

[Q3] I think it is impressive that SafeDICE does not require a hyper-parameter search. Though I agree that (15) is a beautiful and useful equation, do we have some chance to obtain a better $\alpha$?

**Limitations:**

- Unless I miss something, limitations have not adequately addressed.

---

> ### Author Rebuttal · Authors · 2023-08-09
>
> Thank you for your constructive feedback and comments. Please feel free to ask any additional follow-up questions.
>
> **[Responses to Weaknesses]**
>
> **(mathematical contributions)**
> The mathematical contribution of our paper is to present a well-defined mathematical formulation for a new problem setting that cannot be solved with existing DICE-based methods.
>
> First, our proposed algorithm deals with a new problem setting that aims to learn a safe policy using non-preferred demonstrations, and cannot be simply formulated and solved with existing methods including DICE-based algorithms and imitation learning algorithms, which assume that preferred demonstrations are explicitly given.
>
> Secondly, the most important mathematical contribution of this paper is that hyperparameter $\alpha$ can be theoretically determined, unlike existing algorithms in which hyperparameter search is essential and hyperparameter sensitive. Moreover, it is impossible for our algorithm SafeDICE to be formulated and optimized the objective without hyperparameter $\alpha$ selection. Considering the value of $r(s,a)$ in Eq (10) with arbitrary $\alpha$ value, the value inside the log function can be negative for some $\alpha$. However, our proposed $\alpha$ selection rule not only theoretically bounds the difference between the stationary distributions of learned policy and preferred policy, but also guarantees that the value in the log function of equation (10) is always positive (see lines 179-180 and Appendix A.3).
>
> **(Normalized return & SafeRL algorithm)**
> SafeDICE not only outperforms the other baseline algorithms but also the performance itself seems not low. We used a normalized return to improve readability, and the normalized return is a value normalized by optimal performance learned with online constrained RL as follows: $\text{normalized return} = \frac{\text{return}}{\text{maximum return of optimal policy}}$ (i.e. A normalized return value close to 100 means that the tasks are successfully solved). The policy used to generate the preferred demonstration is the policy learned with online constrained RL, and the information of the preferred demonstration shown in Tables 1 and 3 (Appendix D.1 and D.2) is the optimal performance of the online safe RL algorithm (i.e. oracle agent). We will add a clear explanation about normalized return, and the results of this oracle agent as dashed lines to the plots in the final version of the paper.
>
> **[Responses to Questions]**
>
> **(Q1) (Normalized return)**
> As previously mentioned, the normalized return is a value normalized by optimal performance learned with online constrained RL as follows: $\text{normalized return} = \frac{\text{return}}{\text{maximum return of optimal policy}}$ A normalized return value close to 100 means that the tasks are successfully solved and we confirm the tasks are successfully solved.
>
> **(Q2) (Threshold of the Safety Gym)**
> If it is correct that the threshold means the cost limit for each task, we used the default parameter ($d=25$) as in the original paper of the Safety Gym environment (page 16 in [1]). We will add these experimental details to Table 3 (Appendix D.2) in the final version of the paper.
>
> **(Q3) (Chance to obtain a better $\alpha$)**
> Based on Proposition 3.2 of our paper, we can theoretical guarantees on the error bound of the estimated stationary distribution with $\alpha$ selected by Eq (15). As mentioned in the paper (lines 175-178 and Eq (15)), this error is close to zero in most of the real-world scenarios where the behavior of the non-preferred and preferred policies are different. However, since $\epsilon$ is not exactly zero, theoretically there is a small possibility of finding a better $\alpha$.
>
> Practically, the results of Figures 10 and 11 (Appendix E.4) show that it is very difficult to find better $\alpha$ through hyperparameter search.

---

> > ### Comment · Reviewer_Ajjv · 2023-08-11
> > **Thank you for clarification.**
> >
> > Thank you for the addressing my comments at the time of initial review. I read the authors' rebuttal and other reviews. I still consider that this paper is well-written and the proposed method is technically sound. I will keep the original score.

---

> > > ### Author Response · Authors · 2023-08-18
> > > **Response to Reviewer Ajjv**
> > >
> > > Thank you very much for acknowledging our rebuttal. To address the suggestion of the reviewer to compare with safe RL algorithm as oracle agent, we conducted additional experiments for an offline constrained RL method. We run COptiDICE [1], the state-of-the-art offline constrained RL algorithm, using the datasets that are **augmented with additional (ground-truth) reward/cost annotations**. The results are summarized as follows:
> > >
> > >
> > > |[RWRL-Cartpole]|Normalized Return|Average Cost| Cost Violation |CVaR 10% Cost|
> > > |:--------------:|:-----------------:|:---------------:|:--------------:|:---------------:|
> > > |BC| 100.24 $\pm$ 0.40 | 184.44 $\pm$ 0.94 |0.48 $\pm$ 0.00 | 392.75 $\pm$ 2.72 |
> > > |DWBC|99.78 $\pm$ 0.55| 189.97 $\pm$ 3.21 |0.47 $\pm$ 0.00 | 408.71 $\pm$ 4.01 |
> > > |PPL|83.70 $\pm$ 0.87| 229.80 $\pm$ 1.64 |0.48 $\pm$ 0.01 | 440.68 $\pm$ 3.49 |
> > > |DExperts|100.28 $\pm$ 0.38| 186.15 $\pm$ 2.29 |0.48 $\pm$ 0.01 | 397.05 $\pm$ 3.11 |
> > > | SafeDICE |99.91 $\pm$ 0.60| 154.88 $\pm$ 1.79 |0.34 $\pm$ 0.00 | 404.22 $\pm$ 1.56 |
> > > |COptiDICE|**107.95 $\pm$ 2.10**|**97.59 $\pm$ 7.21** |**0.06 $\pm$ 0.01** | **183.63 $\pm$ 6.18** |
> > >
> > >
> > > |[RWRL-Walker]|Normalized Return|Average Cost| Cost Violation |CVaR 10% Cost|
> > > |:--------------:|:-----------------:|:---------------:|:--------------:|:---------------:|
> > > |BC| 98.41 $\pm$ 0.11 | 152.25 $\pm$ 5.33 |0.22 $\pm$ 0.01 | 531.07 $\pm$ 7.53 |
> > > |DWBC|98.99 $\pm$ 0.10| 105.63 $\pm$ 3.85 |0.11 $\pm$ 0.01 | 378.01 $\pm$ 20.88 |
> > > |PPL|99.02 $\pm$ 0.10| 128.46 $\pm$ 4.84 |0.17 $\pm$ 0.01 | 478.75 $\pm$ 12.53 |
> > > |DExperts|95.21 $\pm$ 0.75| 173.20 $\pm$ 11.22 |0.27 $\pm$ 0.03 | 518.38 $\pm$ 5.25 |
> > > | SafeDICE |**99.67 $\pm$ 0.13**| **68.90 $\pm$ 0.83** |**0.00 $\pm$ 0.00** | **124.54 $\pm$ 2.56** |
> > > |COptiDICE|98.07 $\pm$ 0.34|77.86 $\pm$ 2.14 |0.01 $\pm$ 0.00 | 159.91 $\pm$ 8.90 |
> > >
> > >
> > > |[RWRL-Quadruped]|Normalized Return|Average Cost| Cost Violation |CVaR 10% Cost|
> > > |:--------------:|:-----------------:|:---------------:|:--------------:|:---------------:|
> > > |BC| **100.19 $\pm$ 0.08** | 178.10 $\pm$ 5.20 |0.38 $\pm$ 0.02 | 328.10 $\pm$ 1.83 |
> > > |DWBC|99.87 $\pm$ 0.16| 167.15 $\pm$ 3.32 |0.34 $\pm$ 0.02 | 316.86 $\pm$ 1.74 |
> > > |PPL|99.87 $\pm$ 0.09| 169.93 $\pm$ 5.06 |0.35 $\pm$ 0.03 | 319.77 $\pm$ 1.70 |
> > > |DExperts|97.05 $\pm$ 2.62| 178.75 $\pm$ 4.78 |0.38 $\pm$ 0.03 | 317.55 $\pm$ 4.40 |
> > > | SafeDICE |99.68 $\pm$ 0.25| 148.21 $\pm$ 2.84 |0.24 $\pm$ 0.01 | 308.71 $\pm$ 2.90 |
> > > |COptiDICE|87.37 $\pm$ 3.19|**130.21 $\pm$ 5.20** |**0.16 $\pm$ 0.03** | **263.41 $\pm$ 17.00** |
> > >
> > > |[SafetyGym-Goal]|Normalized Return|Average Cost| Cost Violation |CVaR 10% Cost|
> > > |:--------------:|:-----------------:|:---------------:|:--------------:|:---------------:|
> > > |BC| 94.08 $\pm$ 0.60 | 51.47 $\pm$ 1.49 |0.38 $\pm$ 0.02 | 109.66 $\pm$ 1.63 |
> > > |DWBC|93.93 $\pm$ 0.53| 47.96 $\pm$ 1.77 |0.31 $\pm$ 0.02 | 108.79 $\pm$ 2.12 |
> > > |PPL|94.43 $\pm$ 0.66| 52.31 $\pm$ 1.52 |0.37 $\pm$ 0.02 | 111.41 $\pm$ 2.37 |
> > > |DExperts|87.76 $\pm$ 4.41| 50.82 $\pm$ 1.55 |0.36 $\pm$ 0.02 | 112.62 $\pm$ 1.54 |
> > > | SafeDICE |92.04 $\pm$ 0.44| **39.49 $\pm$ 0.94** |**0.21 $\pm$ 0.02** | **93.35 $\pm$ 1.45** |
> > > |COptiDICE|92.98 $\pm$ 1.13|48.79 $\pm$ 0.80 |0.32 $\pm$ 0.01 | 107.25 $\pm$ 0.78 |
> > >
> > > |[SafetyGym-Button]|Normalized Return|Average Cost| Cost Violation |CVaR 10% Cost|
> > > |:--------------:|:-----------------:|:---------------:|:--------------:|:---------------:|
> > > |BC| 88.64 $\pm$ 1.08 | 98.18 $\pm$ 2.19 |0.49 $\pm$ 0.01 | 220.87 $\pm$ 3.48 |
> > > |DWBC|85.53 $\pm$ 0.80| 96.31 $\pm$ 2.38 |0.47 $\pm$ 0.02 | 225.49 $\pm$ 3.52 |
> > > |PPL|86.76 $\pm$ 1.06| 99.97 $\pm$ 2.19 |0.50 $\pm$ 0.02 | 229.46 $\pm$ 3.30 |
> > > |DExperts|86.28 $\pm$ 1.35| 97.30 $\pm$ 1.70 |0.49 $\pm$ 0.01 | 225.93 $\pm$ 5.34 |
> > > | SafeDICE |71.48 $\pm$ 1.33| **70.67 $\pm$ 1.61** |**0.30 $\pm$ 0.01** | **185.64 $\pm$ 1.54** |
> > > |COptiDICE|**94.09 $\pm$ 1.68**|111.10 $\pm$ 3.53 |0.58 $\pm$ 0.02 | 238.46 $\pm$ 3.44 |
> > >
> > > All results indicate averages and standard errors over 5 trials. The results show that SafeDICE is competitive with (or even outperforms in some domains) the constrained offline RL algorithm COptiDICE, even though SafeDICE uses only a much smaller amount of annotation information. We will add these results and explanations to the final version of the paper.
> > >
> > > Please let us know if any further questions or concerns come up and we would be happy to clarify them anytime during the discussion period.
> > >
> > > [1] Jongmin Lee et al., COptiDICE: Offline Constrained Reinforcement Learning via Stationary Distribution Correction Estimation, ICLR 2022

---

### Official Review · Reviewer_ERpc · 2023-07-05

**Soundness:** 2 fair
**Presentation:** 2 fair
**Contribution:** 2 fair
**Rating:** 5
**Confidence:** 4

**Summary:**

Learning safe behaviors from a dataset of demonstrations is a challenging problem. The work presents SafeDICE, an algorithm which learns a safe policy using preference-based imitation learning. The method leverages non-preferred demonstrations in the space of stationary distributions in contrast to prior methods which operate in the policy and discriminator space. SafeDICE reduces the constrained optimization problem into a single convex minimization objective by constructing a Lagrangian and eliminating the maximization problem using a closed form solution. The new objective does not require additional hyperparameter search. Empirical performance demonstrates that SafeDICE is competitive to other methods in satisfying cost constraints.

**Strengths:**

* The paper is well motivated and within the scope of safe Imitation Learning.
* The proposed method presents a novel stationary distribution formulation of the Lagrangian constraints which, to the best of my knowledge, has not been observed before.

**Weaknesses:**

* **Choice of Baselines:** My main concern is the choice of baselines used to compare SafeDICE. The paper only compares with naive BC, DWBC, PPL and DExperts. BC and DWBC are simple imitation learning algorithms which were not designed for constrained optimization problems. PPL utilizes a paired ranking scheme based on reward preferences which is tangential to imitation learning. This leaves DExperts as the only suitable baseline. There exist specific constrained optimization methods such as CPO, PPO-Lagrangian, TRPO and LAMBDA albeit for online and model-based settings [1]. Additionally, there exist offline RL methods [2, 3] which demonstrate safe behaviors as a result of their pessimistic nature. Authors should include these as relevant baselines on small tasks or provide a concrete explanation for why generic Imitation Learning baselines are suitable for constraint satisfaction.
* **Dataset Comparison:** Due to the absence of safe offline IL datasets, the work constructs its own dataset based on preferred and non-preferred demonstrations. My concerns are around the design of the dataset. The paper does not provide a detailed breakdown of the dataset and the number of safe and unsafe actions. Tables 1 and 3 show the number of non-preferred demonstrations which are significantly less in comparison to the preferred demonstrations (50 non-preffered demonstrations against 1000 preferred demonstrations). While the authors make an attempt to compare algorithms on different dataset sizes in Figure 8, these comparisons are still well in the low data regime. In my opinion, the dataset is heavily biased and is not reflective of other benchmarks and real-world applications in safety. The paper could evaluate algorithms on different kinds of datasets with different behavior qualities. For instance, datasets could be constructed from replay buffers of trained lagrangian agents perturbed with gaussian noise or a balanced mix of safe and unsafe actions based on constraint satisfaction.
* **Empirical Evaluation:** Experiments do not comprehensively evaluate the efficacy of the reduced convex minimization problem. The central contribution of the paper is estimating the stationary distribution corrections of the safe policy. This could be evaluated by ablating the Lagrangian with a constraint in which the stationary distributions are absent or non-preferred demonstrations are not utilized. In its current form, empirical evaluation does not throw light on the performance or utility of the proposed objective.
* **Clarity of Derivation:** I am having trouble following the main derivation of Lagrangian $\mathcal{L}(w, \nu ; r_{\alpha})$. Specifically, it would be helpful if authors could explain how they reached step 3 to obtain $w(s,a)$ and $r(s,a)$. It might be helpful to cover intermediate steps in the Appendix or provide a short note as footnote for the reader.

**Questions:**

* Can you please provide a concrete explanation for comparing SafeDICE to generic Imitation Learning baselines? Can you compare SafeDICE to relevant constrained optimization baselines such as CPO, PPO-Lagrangian, TRPO and LAMBDA on small toy tasks?
* What is the breakdown of the dataset? How many safe and unsafe actions does the data consist of? Do samples have cost variables as well? Can you please explain the reason for a lower number of non-preferred demonstrations? How does SafeDICE perform when number of non-preferred demonstrations are increased? Can the evaluation accomodate different kind of datasets?
* How does SafeDICE perform in the absence of stationary distribution constraint? How does the performance vary in the absence of preferred/non-preferred demonstrations?
* Can you please explain how $w(s,a)$ and $r(s,a)$ were obtained?

## Minors

* line 9: learns safe -> learns a safe
* line 43: what does DWBC stand for?
* line 49: what is the degeneration issue?
* line 52: learns safe policy -> learns a safe policy
* line 65: I believe the formulation is a Constrained MDP (CMDP) since the problem setting has cost constraints of the form $\mathcal{C}: S \times A \rightarrow \mathbb{C}$
* line 89: Recent offline -> A recent offline
* line 149: optimization of Eq. 7
* Proposition 3.2: what is $\epsilon$?
* line 258: different two -> two different

[1]. As et. al., Constrained Policy Optimization via Bayesian World Models, ICLR 2022.
[2]. Kumar et. al., Conservative Q-Learning for Offline Reinforcement Learning, NeurIPS 2021.
[3]. Bhardhwaj et. al., Conservative Safety Critics for Exploration, ICLR 2021.

**Limitations:**

The paper would benefit from a discussion on limitations. Authors could highlight the gap in reward-cost trade off and the increasing number of non-preferred demonstrations as potential limitations of the proposed approach.

---

> ### Author Rebuttal · Authors · 2023-08-09
>
> Thank you for your constructive feedback and comments. Please feel free to ask any additional follow-up questions.
>
> **[Responses to Weaknesses]**
>
> **(1) (Choice of Baselines)**
> Please note that the setting we consider in the paper is **offline safe imitation learning, not constrained RL**, and **there is no reward and cost information** in the given offline dataset. Therefore, it is impossible to consider this problem as a constrained optimization problem, and also impossible to consider **online/offline constrained RL** algorithms (i.e. CPO, PPO-Lagrangian, TRPO, and LAMBDA) as our baseline methods.
>
> In the problem setting we consider in the paper, we only assume a small number of **trajectory-level labeled** non-preferred demonstrations $D^N$ and unlabeled demonstrations $D^U$ which consist of both preferred and non-preferred demonstrations. Thus, we consider learning a reward function based on preference and using it to perform offline IL/RL. DWBC is one of the representative offline imitation learning algorithms that can leverage preference information from labeled non-preferred demonstrations and unlabeled demonstrations. Moreover, PPL is an offline RL baseline using learned preference-based reward and weighted behavior cloning which is one of the representative offline RL algorithms. The baseline algorithms we consider in the paper are not just simple IL algorithms, but offline IL/RL-based strong baseline algorithms that utilize preference information.
>
> However, if you are simply curious about the performance of the oracle agent (not the comparison as baseline under fair conditions), the information of the preferred demonstration shown in Tables 1 and 3 (Appendix D.1 and D.2) is the optimal performance of the online safe RL algorithm (i.e. oracle agent). To obtain the preferred demonstration of unlabeled demonstration, we trained the policy with online constrained RL, then generate the preferred demonstration.
>
> **(2) (Dataset Comparison)**
> As mentioned in the paper (lines 265-266), please note that our problem setting assumes a small amount of **labeled** preferred demonstrations and relatively large amounts of unlabeled demonstrations consist of both preferred and non-preferred demonstrations. The reason why the number of **labeled** non-preferred demonstrations in the paper is set to be significantly smaller than unlabeled demonstrations is **to consider the experimental setting similar to real-world problems**. In real-world scenarios, unlabeled demonstrations are relatively easy to obtain, but labeled demonstrations are expensive and difficult to obtain.
>
> And, for the **unlabeled** demonstrations used in all experiments, the ratio of preferred demonstrations to non-preferred demonstrations is 1:1 (ex. For the Walker domain, the number of **labeled** non-preferred demonstrations = 10, the number of **unlabeled** non-preferred demonstrations = 1000, and the number of **unlabeled** preferred demonstrations = 1000). Therefore, the datasets we used in the paper are not biased. We will add these details on the configuration of labeled and unlabeled demonstrations to the final version of the paper.
>
> To avoid misunderstanding, the Figure 8 results mentioned in the review are experiments with varying amounts of **labeled** demonstrations to show the impact of **labeled** non-preferred demonstrations.
>
> In order to evaluate in more various configurations of **unlabeled** demonstrations, we also conducted additional experiments for various configurations of unlabeled demonstrations $D^U$ consisting of different ratios of preferred and non-preferred demonstrations ($(|D^U_P|, |D^U_N|) \in \[(1000, 1000), (500, 1000), (1000, 500), (250, 1000), (1000, 250)\]$, where $|D^U_P|$ and $|D^U_N|$ denote the number of preferred and non-preferred demonstrations in $D^U$, respectively). As can be seen from the uploaded PDF (Figure 1), the results show that SafeDICE outperforms baselines in all settings with various configurations of **unlabeled** demonstrations. These results show that SafeDICE can be applied more effectively in various real-world scenarios regardless of the preferred/non-preferred ratio of unlabeled demonstrations. We will add these results and explanations to the final version of the paper.
>
> **(3) (Empirical Evaluation)**
> We conducted additional experiments to evaluate the impact of the stationary distribution constraint and labeled non-preferred demonstrations. As can be seen from the uploaded PDF (Figure 2), the performance is degraded both in the absence of stationary distribution constraints and in the case of not using labeled non-preferred demonstrations. We will add this result and explanation to the final version of the paper.
>
> **(4) (Clarity of Derivation)**
> We added derivation details to the PDF and will add them to the final version of the paper.
>
> **[Responses to Questions]**
>
> **(1) (Baselines of imitation learning and constrained RL)**
> See the response of **(1) (Choice of Baselines)**
>
> **(2.1) (Breakdown of the dataset & number of safe/unsafe actions)**
> See the response of **(2) (Dataset Comparison)**
>
> In the problem setting we consider in the paper, safe (preferred) and unsafe (non-preferred) are not defined at the action-level, but only defined at the trajectory-level.
>
> **(2.2) (Cost variables)**
> Please note that the setting we consider in the paper is **offline safe imitation learning, not constrained RL**, and **there are no reward and cost variables** in the given offline dataset.
>
> **(3) (Absence of stationary distribution constraint and preferred/non-preferred demonstrations)**
> See the response of **(3) (Empirical Evaluation)** and **(2) (Dataset Comparison)**
>
> **(4) (Derivation of $w(s,a)$ and $r(s,a)$)**
> We added derivation details to the PDF and will add them to the final version of the paper.

---

> > ### Comment · Reviewer_ERpc · 2023-08-13
> > **Response to Authors' Comments**
> >
> > I thank the authors for providing a detailed response. After going through authors responses and other reviewers' comments, my concerns regarding choice of baselines and dataset ablations still remain unaddressed.
> >
> > * **Choice of Baselines**- The authors mention that the problem is casted as an offline safe imitation learning setting. To the best my knowledge, all safe imitation/reinforcement learning problems are solved under the CMDP paradigm. This is because the notion of costs and cost violations are the only means for assessing the safety of an autonomous system. With that said, constrained methods such as CPO, PPO-Lagrangian, TRPO and LAMBDA are all safe IL/RL methods which construct similar lagrangian objectives as proposed by SafeDICE. Thus, it is only fair that an offline lagrangian objective be compared to existing lagrangian objectives in literature. Furthermore, the datasets itself are constructed with a constraint satisfaction criterion using SAC and PPO-Lagrangian agents for RWRL and Safety-Gym respectively. If the authors wish to not compare with prior safety methods then the problem should not be casted as a safe IL/RL problem and not evaluated for costs and cost violations. Similarly, datasets must also be constructed using alternate heuristics instead of constraints. In this case, the work would benefit from other metrics that measure the deviation of a policy from preferred behaviors. In any other case, comparison of SafeDICE with existing safety methods utilizing constrained optimization is paramount.
> >
> > * **Dataset Comparison**- The empirical evaluation provides suitable evidence for SafeDICE's ability to learn from limited preferred demonstrations. However, I am still struggling to understand the dataset splits and their variation. The authors mention that unlabeled demonstrations are split in a 1:1 ratio. Furthermore, experiments in Figure 1 (pdf) evaluate SafeDICE for different unlabeled splits. But what is the split of labeled demonstrations? And how is this split varied in ablations of Figure 8? Additionally, what happens if the dataset has a noisy composition (i.e- gaussian noise or inaccurate constraint satisfaction)?

---

> > > ### Author Response · Authors · 2023-08-18
> > > **Response to Reviewer ERpc (1/3)**
> > >
> > > Thank you very much for acknowledging our rebuttal and further comments.
> > >
> > >
> > > **[Choice of Baselines]**
> > > We apologize for confusing our problem settings (safe imitation learning) with CMDP, and we will revise the paper to make it clearer in the final version.
> > >
> > > However, we respectfully disagree with the reviewer that **all safe imitation/reinforcement learning problems are solved under the CMDP paradigm**. The notion of safety in RL is **not** limited to CMDP, and solving CMDP is just one of the diverse ways to consider safety in RL. Rather, Safe RL refers to a broader set of techniques and approaches in RL that aim to ensure that the agent behaves in a safe and reliable manner (e.g. safe exploration strategies, optimization objectives beyond expected return, etc.), and it has been manifested in various forms, including risk-sensitivity [1,2,3], robust MDP [4,5], constrained MDP, and more. For more discussions about safe RL, we refer to the comprehensive survey paper [6]. Similarly, for safe imitation learning, diverse safety criteria have been considered such as risk-sensitivity [7,8], high-confidence performance bounds [9,10], and so on, besides CMDP [11].
> > >
> > >
> > > In this paper, we consider safe imitation learning in an offline setting, where the agent should learn a safe behavior only from the pre-collected demonstrations (i.e. unlabeled demonstrations that contain both preferred and non-preferred demonstrations & labeled non-preferred demonstrations; see Figure 1 in the paper) without interaction with the environment. Since we are considering imitation learning (**not** RL!), we assume that each demonstration $\tau = ( s_0, a_0, s_1, a_1, \ldots s_T, a_T )$ in the datasets does **not** have any per-timestep reward/cost annotation. This problem setting is appealing when the reward/cost design can be challenging. In contrast to the standard imitation learning that aims to mimic the expert/preferred demonstrations [12, 13], we aim to learn a policy that is **negation** of the non-preferred demonstrations, which serve as safety guidelines (e.g. don't do these undesirable behaviors).
> > >
> > >
> > > Therefore, constrained RL algorithms **cannot be considered** as baseline algorithms to be compared **under fair conditions** since they require a much larger amount of information (i.e. reward and cost annotations for every state-action $D = \{ (s,a,r,c,s')_t \}$) to run, whereas SafeDICE only requires only a few rare annotations (e.g. only dozens of labeled non-preferred demonstrations in our experiments).
> > >
> > >
> > > Nevertheless, at the request of the reviewer, we conducted additional experiments for offline constrained RL method. The referred CPO, PPO-Lagrangian, etc. are online algorithms, thus they cannot be directly applied to the offline setting. We instead run COptiDICE [14], the state-of-the-art offline constrained RL algorithm, using the datasets that are **augmented with additional (ground-truth) reward/cost annotations**. The results are summarized in the next comment (see comment **Response to Reviewer ERpc (2/2)**.
> > >
> > >
> > > [1] Howard and Matheson, Risk-sensitive Markov decision processes, Management Science, 1972
> > >
> > > [2] Chow et al., Risk-Sensitive and Robust Decision-Making: a CVaR Optimization Approach, NIPS 2015.
> > >
> > > [3] Urpi et al., Risk-Averse Offline Reinforcement Learning, ICLR 2021.
> > >
> > > [4] Iyengar, Robust dynamic programming, Mathematics of Operations Research 2005.
> > >
> > > [5] Tamar et al., Scaling up robust mdps using function approximation, ICML 2014.
> > >
> > > [6] Garcia and Fernandez, A Comprehensive Survey on Safe Reinforcement Learning, JMLR 2015.
> > >
> > > [7] Majumdar et al., Risk-sensitive inverse reinforcement learning via coherent risk models, Robotics: Science and Systems 2017.
> > >
> > > [8] Lacotte et al., Risk-Sensitive Generative Adversarial Imitation Learning, AISTATS 2019.
> > >
> > > [9] Brown et al., Efficient Probabilistic Performance Bounds for Inverse Reinforcement Learning, AAAI 2018.
> > >
> > > [10] Brown et al., Safe Imitation Learning via Fast Bayesian Reward Inference from Preferences, ICML 2020.
> > >
> > > [11] Malik et al., Inverse Constrained Reinforcement Learning, ICML 2021.
> > >
> > > [12] Geon-Hyeong Kim et al., DemoDICE: Offline Imitation Learning with Supplementary Imperfect Demonstrations, ICLR 2022.
> > >
> > > [13] Haoran Xu et al., Discriminator-Weighted Offline Imitation Learning from Suboptimal Demonstration, ICML 2022.
> > >
> > > [14] Jongmin Lee et al., COptiDICE: Offline Constrained Reinforcement Learning via Stationary Distribution Correction Estimation, ICLR 2022

---

> > > ### Author Response · Authors · 2023-08-18
> > > **Response to Reviewer ERpc (2/3)**
> > >
> > > |[RWRL-Cartpole]|Normalized Return|Average Cost| Cost Violation |CVaR 10% Cost|
> > > |:--------------:|:-----------------:|:---------------:|:--------------:|:---------------:|
> > > |BC| 100.24 $\pm$ 0.40 | 184.44 $\pm$ 0.94 |0.48 $\pm$ 0.00 | 392.75 $\pm$ 2.72 |
> > > |DWBC|99.78 $\pm$ 0.55| 189.97 $\pm$ 3.21 |0.47 $\pm$ 0.00 | 408.71 $\pm$ 4.01 |
> > > |PPL|83.70 $\pm$ 0.87| 229.80 $\pm$ 1.64 |0.48 $\pm$ 0.01 | 440.68 $\pm$ 3.49 |
> > > |DExperts|100.28 $\pm$ 0.38| 186.15 $\pm$ 2.29 |0.48 $\pm$ 0.01 | 397.05 $\pm$ 3.11 |
> > > | SafeDICE |99.91 $\pm$ 0.60| 154.88 $\pm$ 1.79 |0.34 $\pm$ 0.00 | 404.22 $\pm$ 1.56 |
> > > |COptiDICE|**107.95 $\pm$ 2.10**|**97.59 $\pm$ 7.21** |**0.06 $\pm$ 0.01** | **183.63 $\pm$ 6.18** |
> > >
> > > |[RWRL-Walker]|Normalized Return|Average Cost| Cost Violation |CVaR 10% Cost|
> > > |:--------------:|:-----------------:|:---------------:|:--------------:|:---------------:|
> > > |BC| 98.41 $\pm$ 0.11 | 152.25 $\pm$ 5.33 |0.22 $\pm$ 0.01 | 531.07 $\pm$ 7.53 |
> > > |DWBC|98.99 $\pm$ 0.10| 105.63 $\pm$ 3.85 |0.11 $\pm$ 0.01 | 378.01 $\pm$ 20.88 |
> > > |PPL|99.02 $\pm$ 0.10| 128.46 $\pm$ 4.84 |0.17 $\pm$ 0.01 | 478.75 $\pm$ 12.53 |
> > > |DExperts|95.21 $\pm$ 0.75| 173.20 $\pm$ 11.22 |0.27 $\pm$ 0.03 | 518.38 $\pm$ 5.25 |
> > > | SafeDICE |**99.67 $\pm$ 0.13**| **68.90 $\pm$ 0.83** |**0.00 $\pm$ 0.00** | **124.54 $\pm$ 2.56** |
> > > |COptiDICE|98.07 $\pm$ 0.34|77.86 $\pm$ 2.14 |0.01 $\pm$ 0.00 | 159.91 $\pm$ 8.90 |
> > >
> > > |[RWRL-Quadruped]|Normalized Return|Average Cost| Cost Violation |CVaR 10% Cost|
> > > |:--------------:|:-----------------:|:---------------:|:--------------:|:---------------:|
> > > |BC| **100.19 $\pm$ 0.08** | 178.10 $\pm$ 5.20 |0.38 $\pm$ 0.02 | 328.10 $\pm$ 1.83 |
> > > |DWBC|99.87 $\pm$ 0.16| 167.15 $\pm$ 3.32 |0.34 $\pm$ 0.02 | 316.86 $\pm$ 1.74 |
> > > |PPL|99.87 $\pm$ 0.09| 169.93 $\pm$ 5.06 |0.35 $\pm$ 0.03 | 319.77 $\pm$ 1.70 |
> > > |DExperts|97.05 $\pm$ 2.62| 178.75 $\pm$ 4.78 |0.38 $\pm$ 0.03 | 317.55 $\pm$ 4.40 |
> > > | SafeDICE |99.68 $\pm$ 0.25| 148.21 $\pm$ 2.84 |0.24 $\pm$ 0.01 | 308.71 $\pm$ 2.90 |
> > > |COptiDICE|87.37 $\pm$ 3.19|**130.21 $\pm$ 5.20** |**0.16 $\pm$ 0.03** | **263.41 $\pm$ 17.00** |
> > >
> > > |[SafetyGym-Goal]|Normalized Return|Average Cost| Cost Violation |CVaR 10% Cost|
> > > |:--------------:|:-----------------:|:---------------:|:--------------:|:---------------:|
> > > |BC| 94.08 $\pm$ 0.60 | 51.47 $\pm$ 1.49 |0.38 $\pm$ 0.02 | 109.66 $\pm$ 1.63 |
> > > |DWBC|93.93 $\pm$ 0.53| 47.96 $\pm$ 1.77 |0.31 $\pm$ 0.02 | 108.79 $\pm$ 2.12 |
> > > |PPL|94.43 $\pm$ 0.66| 52.31 $\pm$ 1.52 |0.37 $\pm$ 0.02 | 111.41 $\pm$ 2.37 |
> > > |DExperts|87.76 $\pm$ 4.41| 50.82 $\pm$ 1.55 |0.36 $\pm$ 0.02 | 112.62 $\pm$ 1.54 |
> > > | SafeDICE |92.04 $\pm$ 0.44| **39.49 $\pm$ 0.94** |**0.21 $\pm$ 0.02** | **93.35 $\pm$ 1.45** |
> > > |COptiDICE|92.98 $\pm$ 1.13|48.79 $\pm$ 0.80 |0.32 $\pm$ 0.01 | 107.25 $\pm$ 0.78 |
> > >
> > > |[SafetyGym-Button]|Normalized Return|Average Cost| Cost Violation |CVaR 10% Cost|
> > > |:--------------:|:-----------------:|:---------------:|:--------------:|:---------------:|
> > > |BC| 88.64 $\pm$ 1.08 | 98.18 $\pm$ 2.19 |0.49 $\pm$ 0.01 | 220.87 $\pm$ 3.48 |
> > > |DWBC|85.53 $\pm$ 0.80| 96.31 $\pm$ 2.38 |0.47 $\pm$ 0.02 | 225.49 $\pm$ 3.52 |
> > > |PPL|86.76 $\pm$ 1.06| 99.97 $\pm$ 2.19 |0.50 $\pm$ 0.02 | 229.46 $\pm$ 3.30 |
> > > |DExperts|86.28 $\pm$ 1.35| 97.30 $\pm$ 1.70 |0.49 $\pm$ 0.01 | 225.93 $\pm$ 5.34 |
> > > | SafeDICE |71.48 $\pm$ 1.33| **70.67 $\pm$ 1.61** |**0.30 $\pm$ 0.01** | **185.64 $\pm$ 1.54** |
> > > |COptiDICE|**94.09 $\pm$ 1.68**|111.10 $\pm$ 3.53 |0.58 $\pm$ 0.02 | 238.46 $\pm$ 3.44 |
> > >
> > > All results indicate averages and standard errors over 5 trials. The results show that SafeDICE is competitive with (or even outperforms in some domains) the constrained offline RL algorithm COptiDICE, even though SafeDICE uses only a much smaller amount of annotation information. We will add these results and explanations to the final version of the paper.

---

> > > ### Author Response · Authors · 2023-08-18
> > > **Response to Reviewer ERpc (3/3)**
> > >
> > > **[Dataset Comparison]**
> > >
> > > Q) But what is the split of labeled demonstrations? And how is this split varied in ablations of Figure 8?
> > >
> > > To avoid misunderstanding, we emphasize again that the **labeled demonstrations** consist of **only non-preferred demonstrations (without any preferred demonstrations)** as illustrated in Figure 1 of the paper. Likewise, in the ablations of Figure 8, the labeled demonstrations consist only of non-preferred demonstrations.
> > >
> > > Q) Additionally, what happens if the dataset has a noisy composition (i.e- gaussian noise or inaccurate constraint satisfaction)?
> > >
> > > We additionally conducted the experiments on Walker domain with a dataset which has a noisy composition (i.e. gaussian noise). We add gaussian noise $\epsilon \sim \mathcal{N}(0,\sigma)$ ($\sigma \in [0.001, 0.005, 0.01]$) to the states and actions in the offline dataset and run SafeDICE and baseline algorithms. The results are summarized as Table as follows:
> > >
> > > |w/o noise|Normalized Return|Average Cost| Cost Violation |CVaR 10% Cost|
> > > |:--------------:|:-----------------:|:---------------:|:--------------:|:---------------:|
> > > |BC| 98.41 $\pm$ 0.11 | 152.25 $\pm$ 5.33 |0.22 $\pm$ 0.01 | 531.07 $\pm$ 7.53 |
> > > |DWBC|98.99 $\pm$ 0.10| 105.63 $\pm$ 3.85 |0.11 $\pm$ 0.01 | 378.01 $\pm$ 20.88 |
> > > |PPL|99.02 $\pm$ 0.10| 128.46 $\pm$ 4.84 |0.17 $\pm$ 0.01 | 478.75 $\pm$ 12.53 |
> > > |DExperts|95.21 $\pm$ 0.75| 173.20 $\pm$ 11.22 |0.27 $\pm$ 0.03 | 518.38 $\pm$ 5.25 |
> > > | SafeDICE |**99.67 $\pm$ 0.13**| **68.90 $\pm$ 0.83** |**0.00 $\pm$ 0.00** | **124.54 $\pm$ 2.56** |
> > >
> > > |$\epsilon \sim \mathcal{N}(0,001)$|Normalized Return|Average Cost| Cost Violation |CVaR 10% Cost|
> > > |:--------------:|:-----------------:|:---------------:|:--------------:|:---------------:|
> > > |BC| 99.04 $\pm$ 0.20 | 141.38$\pm$4.24 |0.20$\pm$0.01| 515.63$\pm$5.19 |
> > > |DWBC|99.32$\pm$0.15| 105.52$\pm$0.31 |0.11$\pm$0.00| 397.94$\pm$7.22 |
> > > |PPL|99.23$\pm$0.15| 125.94$\pm$3.34 |0.16$\pm$0.01| 476.51$\pm$11.22|
> > > |DExperts|99.15$\pm$0.06| 142.74$\pm$2.08 |0.20$\pm$0.01| 525.37$\pm$4.45|
> > > | SafeDICE |**100.01$\pm$0.12**| **68.14$\pm$0.75** |**0.01$\pm$0.00** | **130.90$\pm$3.10** |
> > >
> > > |$\epsilon \sim \mathcal{N}(0,005)$|Normalized Return|Average Cost| Cost Violation |CVaR 10% Cost|
> > > |:--------------:|:-----------------:|:---------------:|:--------------:|:---------------:|
> > > |BC| 79.86$\pm$0.67| 264.69$\pm$3.76 |0.65$\pm$0.01| 451.98$\pm$5.56 |
> > > |DWBC|80.64$\pm$0.90| 220.49$\pm$2.55 |0.49$\pm$0.01| 388.21$\pm$1.28|
> > > |PPL|80.36$\pm$1.01| 241.55$\pm$4.92 |0.57$\pm$0.02| 417.08$\pm$4.83|
> > > |DExperts|78.35$\pm$0.38| 265.49$\pm$2.49 |0.66$\pm$0.01| 443.93$\pm$3.89|
> > > | SafeDICE |**84.92$\pm$0.37**| **166.36$\pm$5.52** |**0.21$\pm$0.03** | **279.55$\pm$8.43** |
> > >
> > > |$\epsilon \sim \mathcal{N}(0,01)$|Normalized Return|Average Cost| Cost Violation |CVaR 10% Cost|
> > > |:--------------:|:-----------------:|:---------------:|:--------------:|:---------------:|
> > > |BC| 45.87$\pm$1.40| 393.61$\pm$5.98 |**0.98$\pm$0.00**| 487.08$\pm$6.35|
> > > |DWBC|48.340$\pm$1.69| 352.27$\pm$10.08 |0.93$\pm$0.02| 456.19$\pm$9.29|
> > > |PPL|44.79$\pm$1.07| 352.19$\pm$9.14 |0.92$\pm$0.03| 462.84$\pm$3.43|
> > > |DExperts|43.03$\pm$2.01| 376.97$\pm$9.72 |0.95$\pm$0.02| 480.84$\pm$6.94|
> > > | SafeDICE |**58.88$\pm$0.65**| **341.88$\pm$7.27** |0.97$\pm$0.01| **428.15$\pm$6.33** |
> > >
> > > All results indicate averages and standard errors over 5 trials. The results show that the performance of all algorithms deteriorates with the noise of the dataset. However, even when the dataset has a noisy composition, SafeDICE outperforms the other baseline algorithms.
> > >
> > > Please let us know if any further questions or concerns come up and we would be happy to clarify them anytime during the discussion period.

---

> > > > ### Comment · Reviewer_ERpc · 2023-08-18
> > > > **Response to Comments**
> > > >
> > > > Thank you for the response and the additional experiments. Authors have provided additional comparisons on noisy datasets and a constrained offline baseline which addresses my concern. I still encourage the authors to compare SafeDICE with the existing oracle online baseline of PPO-Lagrangian. Note that SafeDICE need not outperform the orcale, but the comparison must include the behavior policy as a means for benchmarking the performance gap.
> > > >
> > > > Since the authors have addressed most of my main concerns around dataset comparison, empirical evaluation, choice of baselines and clarity of derivation, I would like to raise my score. I thank the authors for their response.

---

> > > > > ### Author Response · Authors · 2023-08-21
> > > > >
> > > > > Thank you for your thoughtful response and suggestion for the additional comparison with the oracle online baseline algorithms. The table below shows the performance of the oracle agent in each domain. We will add this result as the performance of oracle agents to the final version of the paper.
> > > > >
> > > > > |Domain|Normalized Return|Average Cost| Cost Violation |CVaR 10% Cost|
> > > > > |:--------------:|:-----------------:|:---------------:|:--------------:|:---------------:|
> > > > > |Cartpole| 97.48 $\pm$ 0.11 | 71.95 $\pm$ 0.16 |0.15 $\pm$ 0.00 | 394.00 $\pm$ 1.30 |
> > > > > |Walker|101.44 $\pm$ 0.09| 55.79 $\pm$ 0.16 |0.04 $\pm$ 0.00 | 207.16 $\pm$ 0.97 |
> > > > > |Quadruped|101.14 $\pm$ 0.00| 96.62 $\pm$ 0.28 |0.00 $\pm$ 0.00 | 125.19 $\pm$ 0.57 |
> > > > > |Goal|99.83 $\pm$ 0.14| 13.91 $\pm$ 0.19 |0.00 $\pm$ 0.00 | 35.39 $\pm$ 0.22 |
> > > > > |Button|38.67 $\pm$ 0.73| 12.32 $\pm$ 0.25 |0.00 $\pm$ 0.00| 41.20 $\pm$ 0.25 |

---

### Official Review · Reviewer_jY8W · 2023-07-08

**Soundness:** 3 good
**Presentation:** 3 good
**Contribution:** 2 fair
**Rating:** 5
**Confidence:** 3

**Summary:**

This paper proposes an offline safe imitation method to avoid non-preferred demonstrations. The mixture coefficient is alpha, and the paper provides an effective way to obtain hyperparameter instead of hyperparameter search techniques. The paper uses scarce but labeled non-preferred demonstrations from the non-preferred policy.

**Strengths:**

The paper models the stationary distributions as a mixture of preferred policy and non-preferred policy. The mixture coefficient is alpha, and the paper provides an effective way to obtain hyperparameter instead of hyperparameter search techniques.

The paper is easy to follow and understand.

Claims made at the end of the introduction are supported by experimental results.

**Weaknesses:**

The problem setup is unclear to me. In the offline imitation learning setting, there's a line of research consider robust imitation learning (such as [1]) to discard non-preferred samples. The authors may need to discuss why labeled non-preferred demonstrations is nececssary in this situation.

Furthermore, more discussion can be added to explain why SafeDICE is not sensitive to the number of labeled non-preferred demonstrations.

[1] Robust Imitation Learning from Corrupted Demonstrations, https://arxiv.org/abs/2201.12594

**Questions:**

Since the additional cost of hyperparameter search techniques have influence on the computational cost rather than final performance, it is not clear why SafeDICE performs better with the  obtained by Eq. 15. Can you give more explanations about the results between SafeDICE and DWBC?

**Limitations:**

As stated above, I would like to compare this methods with those robust imitation learning methods in the offline setting.

---

> ### Author Rebuttal · Authors · 2023-08-09
>
> Thank you for your constructive feedback and comments. Please feel free to ask any additional follow-up questions.
>
> **[Responses to Weaknesses]**
>
> **(1.1) (Problem Setup & The need for labeled non-preferred demonstrations)**
> In our work, we focus on solving safe imitation learning in an offline setting. We assume scarce but **labeled** non-preferred demonstrations $D^N$, and abundant but **unlabeled** demonstrations $D^U$ of both preferred and non-preferred behavior. Our goal is to learn a safe policy that follows the preferred behavior while avoiding the non-preferred behavior.
>
> Please note that our problem setup and the goal are clearly different from robust imitation learning (such as [1]). Robust imitation learning [1] assumes only unlabeled demonstrations and focuses on filtering out **outliers** from unlabeled demonstrations. However, in order to filter out outliers, robust imitation learning [2] assumes that the **majority of the unlabeled demonstrations are preferred demonstrations**.  In other words, if the non-preferred demonstration ratio of unlabeled demonstration is greater than 0.5, it cannot be applied. Therefore, cases in which preferred behavior can be learned only with unlabeled demonstrations without any labeled demonstrations are very limited.
>
> On the other hand, we assume a small number of labeled non-preferred demonstrations, but **no assumptions are needed for unlabeled demonstrations** (ex. the ratios of preferred and non-preferred demonstrations in unlabeled demonstrations). To show that SafeDICE successfully works regardless of the configuration of unlabeled demonstration, we additional conducted experiments for various configurations of unlabeled demonstrations $D^U$ consisting of different ratios of preferred and non-preferred demonstrations  ($(|D^U_P|, |D^U_N|) \in \[(1000, 1000), (500, 1000), (1000, 500), (250, 1000), (1000, 250)\]$, where $|D^U_P|$ and $|D^U_N|$ denote the number of preferred and non-preferred demonstrations in $D^U$, respectively). As can be seen from the uploaded PDF (Figure 1), the results show that SafeDICE outperforms baselines in all settings with various configurations of **unlabeled** demonstrations. These results show that SafeDICE can be applied more effectively in various real-world scenarios regardless of the preferred/non-preferred ratio of unlabeled demonstrations. We will add these results and explanations to the final version of the paper.
>
> **(1.2) (Robust Imitation Learning)**
> As the reviewer pointed out, BCND [2] and RBC [1] also share the aim of learning a safe policy, similar to SafeDICE. However, BCND and RBC focus on **filtering out outliers**, whereas SafeDICE imitates provided demonstrations while excluding non-preferred demonstrations.
>
> Both BCND and RBC aim to learn an optimal policy using only the provided unlabeled demonstrations (unlabeled demonstrations are referred to as noisy demonstrations in [2] and as corrupted demonstrations in [1], respectively). **Assuming that the majority of the unlabeled demonstrations are preferred demonstrations**, BCND and RBC recover an optimal policy via iterative mode-seeking and Median-of-Means (MOM), respectively. While these approaches are effective for filtering out outliers, they might face challenges when preferred demonstrations are not the majority of the unlabeled demonstrations.
>
> In contrast, SafeDICE aims to learn a safe policy that follows the preferred behavior using non-preferred demonstrations and unlabeled demonstrations. We want to emphasize that unlabeled demonstrations could include over half of the non-preferred demonstrations (see the results in the uploaded PDF, Figure 1).
>
> We will add this discussion on robust imitation learning to the related work section of the final version of the paper.
>
> [1] Liu, Liu, et al. "Robust imitation learning from corrupted demonstrations." arXiv 2022.
> [2] Sasaki, Fumihiro, and Ryota Yamashina. "Behavioral cloning from noisy demonstrations." ICLR 2021.
>
> **(2) (Sensitive to the number of labeled non-preferred demonstrations)**
> In the case of preference-based reward learning (i.e. reward learning in PPL), since trajectory-level samples are used for reward learning (see Eq (40) in Appendix C.3), it is relatively sensitive to the number of labeled non-preferred demonstrations $|D^N|$. On the other hand, DWBC and SafeDICE use state-action-pair-level samples for reward learning, so they learn relatively sample-efficiently and are not sensitive to $|D^N|$. For example, if $|D^N|=10$ and tiemstep_of_trajectory=1000, only 10 non-preferred samples are used in reward learning for PPL,  but in the case of DWBC and SafeDICE, 10000 non-preferred samples are used.
>
> **[Responses to Questions]**
>
> **(Related to hyperparameter search)**
> In the offline imitation learning setting where further environment interactions are not available, hyperparameter search through evaluation is not possible. Therefore, in offline imitation learning, it is very important to work without hyperparameter search. And as can be seen from the results in Appendix E.2 (Figures 6 and 7), baseline algorithms are hyperparameter sensitive, and finding the optimal hyperparameter is very challenging and expensive.
>
> **(Comparison with DWBC)**
> As shown in Appendix E.2 (Figures 6 and 7), since DWBC is hyperparameter sensitive, finding the optimal hyperparameter is difficult even if the hyperparameter search is allowed.
>
> In addition to the hyperparameter-free or not, the main difference between SafeDICE and DWBC is that SafeDICE essentially leverages the non-preferred demonstrations **in the space of stationary distributions (i.e. directly estimates the stationary distribution corrections of the policy that imitate the demonstrations excluding the non-preferred behavior)**, unlike DWBC that leverages it in the space of policy or the learning process of the discriminator.

---

> > ### Comment · Reviewer_jY8W · 2023-08-21
> > **Response to Authors' Comments**
> >
> > Thanks for the detailed reply. The authors' dicussion on robust imitation learning and the comparisons to [1] and [2] is convincing, and weakens my concern. I suggest adding this part to the final version of the paper, which would be very useful for the readers. And I will update the scores accordingly.

---

### Official Review · Reviewer_kwD7 · 2023-07-08

**Soundness:** 4 excellent
**Presentation:** 4 excellent
**Contribution:** 3 good
**Rating:** 7
**Confidence:** 4

**Summary:**

This paper focuses on offline safe imitation learning (IL) setting. There are several unique properties of the problem setting: 1) there exists non-preferred (constraint violated) demonstrations, 2) there exists massive unlabelled data where you don't know whether they are preferred or non-preferred, 3) you don't have reward function and interactive environment.

The proposed SafeDICE method builds upon DIstribution Correction Estimation (DICE), but addresses several challenges. The first is how to takes non-preferred demonstrations into account? To do so, we assume an underlying unknown mix ratio alpha and the unlabelled dataset is the mixed dataset of non-preferred and preferred demonstrations. Doing such assumption allows us to follow DICE workflow. At the same time, we can use a preference-based reward function to estimate the stationary distribution d(s, a) of a state-action pair. The reward function is similar to those used in inverse RL and can be achieved by a classifier discriminating whether a datapoint is from non-preferred dataset or not. The authors derives a closed-form solution to avoid the training instability and provide an approximation to the unknown mix ratio to avoid hyperparameter search.

Experiments show the proposed method can achieve good performance while greatly reduce the constraint violation in Real-World RL suite and SafetyGym.

**Strengths:**

1. The paper is well-written and easy to follow.
2. The proposed method is solid, clear and looks sound to me.
3. The problem setting is novel and important.

**Weaknesses:**

1. The experiment showcases the superior performance of proposed method. But more can be added to study the particular behavior of the proposed method in different settings, such as when you have lot of / scare non-preferred demonstrations.


**Questions:**

1. Some ablation studies should be added to support the claim. For example, what if we don't use the approximation of mix ratio alpha but really do a hyperparameter search?
2. Can we have an experiment/discussion section discuss the impact of labelled non-preferred data? Like, if we have plenty of them and if we have very few those data, what would happen?
3. Can we compare/discuss to offline RL method where you can have reward function? In offline RL the data is not always preferred as the author discussed for those offline IL method. So I wonder offline RL shares some settings in current problem. When comparing online Safe RL method, a competitive baseline is to use online RL with reward shaping (the constraint violation can be considered as a negative reward). Would this method also be competitive in offline Safe RL setting?

**Limitations:**

It seems that there is no much effort put in the paper to discuss the limitations. One I can think of is the collection of non-preferred data is in some sense still dangerous. The "non-preferred data efficiency" is not discussed in the paper yet. That is, how efficient the proposed method can learn from how much non-preferred data. This is also important since as the authors said collecting those data is costly.

---

> ### Author Rebuttal · Authors · 2023-08-09
>
> Thank you for your constructive feedback and comments. Please feel free to ask any additional follow-up questions.
>
> **[Responses to Weaknesses]**
>
> **(1) (Different settings with non-preferred demonstrations)**
> Thank you for your suggestion for the additional experiment varying the amount of **labeled** non-preferred demonstrations. Our paper already provides the experimental results you suggested in Section 5.1 (lines 284-295) and Appendix E.3 (Figures 8 and 9). As can be seen from the results, as the number of labeled non-preferred demonstrations $|D^N|$ decreases, SafeDICE shows a greater difference from other baseline algorithms, significantly outperforming.
> In order to evaluate our algorithm with more various configurations of **unlabeled** demonstrations, we also conducted additional experiments for various configurations of unlabeled demonstrations $D^U$ consisting of different ratios of preferred and non-preferred demonstrations ($(|D^U_P|, |D^U_N|) \in \[ (1000, 1000), (500, 1000), (1000, 500), (250, 1000), (1000, 250)\]$, where $|D^U_P|$ and $|D^U_N|$ denote the number of preferred and non-preferred demonstrations in $D^U$, respectively). As can be seen from the uploaded PDF (Figure 1), the results show that SafeDICE outperforms baselines in all settings with various configurations of **unlabeled** demonstrations. These results show that SafeDICE can be applied more effectively in various real-world scenarios regardless of the preferred/non-preferred ratio of unlabeled demonstrations. We will add these results and explanations to the final version of the paper.
>
> **[Responses to Questions]**
>
> **(1) (Experiment with hyperparameter search for $\alpha$)**
> Our paper already provides a comparison with the results of performing a hyperparameter search in Appendix E.4 (Figures 10 and 11). As shown in Figures 10 and 11, the results show various performances depending on the value of $\alpha$, and the result with $\alpha$ value selected by Eq. (15) shows the best performance.
>
> **(2) (Impact of labeled non-preferred data)**
> As mentioned in (Responses to Weaknesses 1), our paper already provides the experimental results you suggested in Section 5.1 (lines 284-295) and Appendix E.3 (Figures 8 and 9). We observed that, for all algorithms, the performance in terms of safety gradually degrades as the number of labeled non-preferred demonstrations $|D^N|$ decreases. However, as $|D^N|$ decreases, SafeDICE shows a greater difference from other baseline algorithms, significantly outperforming.
>
> **(3) (Comparison with offline Safe RL)**
> Please note that the offline imitation learning setting we consider in the paper does **not assume labels for both the reward and cost**. We assume only a small number of trajectory-level labeled preferred demonstrations and unlabeled demonstrations. Therefore, it is impossible to directly apply the offline safe RL algorithms. However, we can consider learning a reward function based on preference and then using it to perform offline RL. In the paper, PPL is an offline RL baseline using learned preference-based reward and weighted behavior cloning which is one of the representative offline RL algorithms.

---

> > ### Comment · Reviewer_kwD7 · 2023-08-15
> >
> > Thanks for the response. My score remains unchanged. Even though you "do not assume labels for both the reward and cost", you can still run the experiment with some sorts of reward and cost as safety gym indeed provides these.

---

> > > ### Author Response · Authors · 2023-08-18
> > > **Response to Reviewer kwD7**
> > >
> > > Thank you very much for acknowledging our rebuttal and suggestion for comparison with offline constrained RL algorithms. At the request of the reviewer, we conducted additional experiments for the offline constrained RL method. We run COptiDICE [1], the state-of-the-art offline constrained RL algorithm, using the datasets that are **augmented with additional (ground-truth) reward/cost annotations**. Please note that, in this experiment, constrained RL algorithm COptiDICE **cannot be considered** as baseline algorithms to be compared **under fair conditions** since they require a much larger amount of information (i.e. reward and cost annotations for every state-action $D = \{ (s,a,r,c,s')_t \}$) to run, whereas SafeDICE only requires only a few rare annotations (e.g. only dozens of labeled non-preferred demonstrations in our experiments).
> > >
> > > The results are summarized as follows:
> > >
> > > |[RWRL-Cartpole]|Normalized Return|Average Cost| Cost Violation |CVaR 10% Cost|
> > > |:--------------:|:-----------------:|:---------------:|:--------------:|:---------------:|
> > > |BC| 100.24 $\pm$ 0.40 | 184.44 $\pm$ 0.94 |0.48 $\pm$ 0.00 | 392.75 $\pm$ 2.72 |
> > > |DWBC|99.78 $\pm$ 0.55| 189.97 $\pm$ 3.21 |0.47 $\pm$ 0.00 | 408.71 $\pm$ 4.01 |
> > > |PPL|83.70 $\pm$ 0.87| 229.80 $\pm$ 1.64 |0.48 $\pm$ 0.01 | 440.68 $\pm$ 3.49 |
> > > |DExperts|100.28 $\pm$ 0.38| 186.15 $\pm$ 2.29 |0.48 $\pm$ 0.01 | 397.05 $\pm$ 3.11 |
> > > | SafeDICE |99.91 $\pm$ 0.60| 154.88 $\pm$ 1.79 |0.34 $\pm$ 0.00 | 404.22 $\pm$ 1.56 |
> > > |COptiDICE|**107.95 $\pm$ 2.10**|**97.59 $\pm$ 7.21** |**0.06 $\pm$ 0.01** | **183.63 $\pm$ 6.18** |
> > >
> > > |[RWRL-Walker]|Normalized Return|Average Cost| Cost Violation |CVaR 10% Cost|
> > > |:--------------:|:-----------------:|:---------------:|:--------------:|:---------------:|
> > > |BC| 98.41 $\pm$ 0.11 | 152.25 $\pm$ 5.33 |0.22 $\pm$ 0.01 | 531.07 $\pm$ 7.53 |
> > > |DWBC|98.99 $\pm$ 0.10| 105.63 $\pm$ 3.85 |0.11 $\pm$ 0.01 | 378.01 $\pm$ 20.88 |
> > > |PPL|99.02 $\pm$ 0.10| 128.46 $\pm$ 4.84 |0.17 $\pm$ 0.01 | 478.75 $\pm$ 12.53 |
> > > |DExperts|95.21 $\pm$ 0.75| 173.20 $\pm$ 11.22 |0.27 $\pm$ 0.03 | 518.38 $\pm$ 5.25 |
> > > | SafeDICE |**99.67 $\pm$ 0.13**| **68.90 $\pm$ 0.83** |**0.00 $\pm$ 0.00** | **124.54 $\pm$ 2.56** |
> > > |COptiDICE|98.07 $\pm$ 0.34|77.86 $\pm$ 2.14 |0.01 $\pm$ 0.00 | 159.91 $\pm$ 8.90 |
> > >
> > > |[RWRL-Quadruped]|Normalized Return|Average Cost| Cost Violation |CVaR 10% Cost|
> > > |:--------------:|:-----------------:|:---------------:|:--------------:|:---------------:|
> > > |BC| **100.19 $\pm$ 0.08** | 178.10 $\pm$ 5.20 |0.38 $\pm$ 0.02 | 328.10 $\pm$ 1.83 |
> > > |DWBC|99.87 $\pm$ 0.16| 167.15 $\pm$ 3.32 |0.34 $\pm$ 0.02 | 316.86 $\pm$ 1.74 |
> > > |PPL|99.87 $\pm$ 0.09| 169.93 $\pm$ 5.06 |0.35 $\pm$ 0.03 | 319.77 $\pm$ 1.70 |
> > > |DExperts|97.05 $\pm$ 2.62| 178.75 $\pm$ 4.78 |0.38 $\pm$ 0.03 | 317.55 $\pm$ 4.40 |
> > > | SafeDICE |99.68 $\pm$ 0.25| 148.21 $\pm$ 2.84 |0.24 $\pm$ 0.01 | 308.71 $\pm$ 2.90 |
> > > |COptiDICE|87.37 $\pm$ 3.19|**130.21 $\pm$ 5.20** |**0.16 $\pm$ 0.03** | **263.41 $\pm$ 17.00** |
> > >
> > > |[SafetyGym-Goal]|Normalized Return|Average Cost| Cost Violation |CVaR 10% Cost|
> > > |:--------------:|:-----------------:|:---------------:|:--------------:|:---------------:|
> > > |BC| 94.08 $\pm$ 0.60 | 51.47 $\pm$ 1.49 |0.38 $\pm$ 0.02 | 109.66 $\pm$ 1.63 |
> > > |DWBC|93.93 $\pm$ 0.53| 47.96 $\pm$ 1.77 |0.31 $\pm$ 0.02 | 108.79 $\pm$ 2.12 |
> > > |PPL|94.43 $\pm$ 0.66| 52.31 $\pm$ 1.52 |0.37 $\pm$ 0.02 | 111.41 $\pm$ 2.37 |
> > > |DExperts|87.76 $\pm$ 4.41| 50.82 $\pm$ 1.55 |0.36 $\pm$ 0.02 | 112.62 $\pm$ 1.54 |
> > > | SafeDICE |92.04 $\pm$ 0.44| **39.49 $\pm$ 0.94** |**0.21 $\pm$ 0.02** | **93.35 $\pm$ 1.45** |
> > > |COptiDICE|92.98 $\pm$ 1.13|48.79 $\pm$ 0.80 |0.32 $\pm$ 0.01 | 107.25 $\pm$ 0.78 |
> > >
> > > |[SafetyGym-Button]|Normalized Return|Average Cost| Cost Violation |CVaR 10% Cost|
> > > |:--------------:|:-----------------:|:---------------:|:--------------:|:---------------:|
> > > |BC| 88.64 $\pm$ 1.08 | 98.18 $\pm$ 2.19 |0.49 $\pm$ 0.01 | 220.87 $\pm$ 3.48 |
> > > |DWBC|85.53 $\pm$ 0.80| 96.31 $\pm$ 2.38 |0.47 $\pm$ 0.02 | 225.49 $\pm$ 3.52 |
> > > |PPL|86.76 $\pm$ 1.06| 99.97 $\pm$ 2.19 |0.50 $\pm$ 0.02 | 229.46 $\pm$ 3.30 |
> > > |DExperts|86.28 $\pm$ 1.35| 97.30 $\pm$ 1.70 |0.49 $\pm$ 0.01 | 225.93 $\pm$ 5.34 |
> > > | SafeDICE |71.48 $\pm$ 1.33| **70.67 $\pm$ 1.61** |**0.30 $\pm$ 0.01** | **185.64 $\pm$ 1.54** |
> > > |COptiDICE|**94.09 $\pm$ 1.68**|111.10 $\pm$ 3.53 |0.58 $\pm$ 0.02 | 238.46 $\pm$ 3.44 |
> > >
> > > All results indicate averages and standard errors over 5 trials. The results show that SafeDICE is competitive with (or even outperforms in some domains) the constrained offline RL algorithm COptiDICE, even though SafeDICE uses only a much smaller amount of annotation information. We will add these results and explanations to the final version of the paper.
> > >
> > > Please let us know if any further questions or concerns come up and we would be happy to clarify them anytime during the discussion period.
> > >
> > > [1] Jongmin Lee et al., COptiDICE: Offline Constrained Reinforcement Learning via Stationary Distribution Correction Estimation, ICLR 2022

---

### Author Rebuttal · Authors · 2023-08-09

We thank all the reviewers for their constructive feedback and comments. Below we restate the main clarification and experiments of our rebuttal. If you have any additional questions or concerns to our response, we are happy to provide additional responses during the rebuttal period.

**[Clarification of problem settings and baselines]**
Please note that the setting we consider in the paper is **offline safe imitation learning, not constrained RL**, and **there is no reward and cost information in the given offline dataset**. We only assume a small number of *trajectory-level* **labeled** preferred demonstrations and abundant **unlabeled** demonstrations. Therefore, it is impossible to consider this problem as a constrained optimization problem, and also **impossible to consider online/offline constrained RL** algorithms as our baseline methods. However, the baseline algorithms we consider in the paper are not just simple IL algorithms, but offline IL/RL-based strong baseline algorithms that **can utilize preference information**.

**[Additional experiments]**
1. In order to evaluate in more various configurations of unlabeled demonstrations, we also conducted additional experiments for various configurations of unlabeled demonstrations $D^U$ consisting of different ratios of preferred and non-preferred demonstrations ($(|D^U_P|, |D^U_N|) \in \[(1000, 1000), (500, 1000), (1000, 500), (250, 1000), (1000, 250)\]$, where $|D^U_P|$ and $|D^U_N|$ denote the number of preferred and non-preferred demonstrations in $D^U$, respectively). As can be seen from the uploaded PDF (Figure 1), the results show that SafeDICE outperforms baselines in all settings with various configurations of unlabeled demonstrations. These results show that SafeDICE can be applied more effectively in various real-world scenarios regardless of the preferred/non-preferred ratio of unlabeled demonstrations. We will add these results and explanations to the final version of the paper.

2. We conducted additional experiments to evaluate the impact of the stationary distribution constraint and labeled non-preferred demonstrations. As can be seen from the uploaded PDF (Figure 2), the performance is degraded both in the absence of stationary distribution constraints and in the case of not using labeled non-preferred demonstrations. We will add this result and explanation to the final version of the paper.

---

### Decision · Program_Chairs · 2023-09-21

**Decision:**

Accept (poster)

**Comment:**

The authors address the problem of safe imitation learning in a problem formulation that assumes that, in addition to demonstrated target behavior, a small set of "non-preferred demonstrations" are provided that demonstrate the types of behaviors to avoid. A novel approach, SafeDICE, is introduced and empirical results demonstrate its effectiveness.

Initial reviews showed a large difference in assessments between reviewers. Noted strengths included the novel and relevant problem setting, soundness and novelty of the approach, strong empirical results and overall clarity of the presentation.

Noted weaknesses included the depth of empirical evaluation to go beyond purely proving strong performance of the approach and to provide deeper insights on e.g., the role of non-preferred demonstrations and other components of the algorithm as well as comparison to oracle and additional baseline performance. In addition, opportunities for improving the presentation were identified.

During the rebuttal period, reviewers and authors engaged in detailed conversations, clarified open questions, and agreed on additional oracle and other results to be included in the main paper. After this discussion, reviewers now unanimously recommend acceptance. The authors are strongly encouraged to consider all feedback when preparing the camera ready version.